



# Evaluation of the 15-year ROM SAF monthly mean GPS radio occultation climate data record

Hans Gleisner[1], Kent B. Lauritsen[1], Johannes K. Nielsen[1], and Stig Syndergaard[1]

[1]Danish Meteorological Institute, Lyngbyvej 100, Copenhagen, Denmark

**Correspondence:** Hans Gleisner (hgl@dmi.dk)

**Abstract.** We here present results from an evaluation of the ROM SAF gridded monthly-mean climate data record (CDR v1.0), based on GPS radio occultation (RO) data from the CHAMP, GRACE, COSMIC, and Metop satellite missions. Systematic differences between RO missions, as well as differences of RO data relative to ERA-Interim reanalysis data, are quantified. The methods used to generate gridded monthly mean data are described, and the correction of monthly-mean RO climatologies for
sampling errors, which is essential for combining data from RO missions with different sampling characteristics, is evaluated.

We find a good overall agreement between the ROM SAF gridded monthly-mean CDR and the ERA-Interim reanalysis, particularly in the 8-30 km height interval. Here, the differences largely reflect time-varying biases in ERA-Interim, suggesting that the RO data record has a better long-term stability than ERA-Interim. Above 30-40 km altitude, the differences are larger, particularly for the pre-COSMIC era.

In the 8–30 km altitude region, the observational data record exhibits a high degree of internal consistency between the RO satellite missions, allowing us to combine data into multi-mission records. For global mean bending angle the consistency is better than 0.04%, for refractivity 0.05%, and for global mean dry temperature the consistency is better than 0.15 K in this height interval. At altitudes up to 40 km, these numbers increase to 0.08%, 0.11%, and 0.50 K, respectively. The numbers can be up to a factor of 2 larger for certain latitude bands compared to global means. Below about 6-8 km the RO mission
differences are larger, reducing the possibilities to generate multi-mission data records. We also find that the residual sampling errors are about one third of the original and that they include a component most likely related to diurnal or semi-diurnal cycles.

## 1 Introduction

Radio Occultation (RO) measurements, exploiting radio signals emitted by Global Navigation Satellite System (GNSS) satel-
lites, are increasingly making important contributions to the global observing system. RO data now have a significant impact in weather forecasting (e.g., *Healy*, 2005; *Cardinali and Healy*, 2014) and in atmospheric reanalysis (*Poli et al*, 2010; *Simmons et al.*, 2017), and as the RO data records become longer, they are also increasingly useful for climate monitoring and climate studies (e.g., *Steiner et al.*, 2011; *Anthes*, 2011). The RO measurement technique has a number of attractive features: it provides





geophysical information with high vertical resolution throughout the troposphere and stratosphere, it is insensitive to clouds and the underlying surface, and it has an intrinsic long-term stability that does not rely on inter-calibration between satellites or instruments (*Kursinski et al.*, 1997; *Leroy et al.*, 2006). The latter feature is particularly important for climate applications where small differences in atmospheric properties that develops over decades are monitored.

The RO Meteorology Satellite Application Facility (ROM SAF), which is a decentralized operational RO processing center under EUMETSAT, has recently undertaken a reprocessing of RO data from four satellite missions: CHAMP (*Wickert et al.*, 2001), GRACE (*Beyerle et al.*, 2005), COSMIC (*Anthes et al.*, 2008), and Metop (*Luntama et al.*, 2008). The reprocessed RO data cover the time period from late 2001 to end of 2016. Over the last few years, reprocessing activities including these four RO missions, or subsets of them, have been undertaken by several processing centers in a joint effort to quantify the structural
uncertainty of RO data. The results have been described in *Ho et al.* (2009, 2012) and *Steiner et al.* (2013), where the impacts of using different processing schemes were investigated. At low- and mid-latitudes between 8 and 25 km the associated structural uncertainties were found to be small enough for RO data to be used in climate change detection studies. For higher altitudes and latitudes, there are mission-specific limitations that need to be considered.

     RO data from the four satellite missions included in the ROM SAF reprocessing have somewhat different characteristics
related to instrumental noise, signal tracking methods and accuracies, data numbers, and spatio-temporal sampling characteristics. The low-level data (excess phase, amplitude, and satellite orbit data) may also exhibit subtle differences depending on the source of those data. If such differences propagate to the retrieved geophysical monthly-mean data, and are large enough, the evolution of the global RO constellation may lead to spurious long-term variability as new satellite missions replace older ones. However, despite differences between the RO missions there is an expectation that measurements from different missions
can be combined without any adjustments or inter-calibrations to form long time series of RO data, provided that they use the same processing scheme (e.g., *Foelsche et al.*, 2011; *Angerer et al.*, 2017). Multi-mission RO time series have been used in several studies, implicitly assuming inter-mission consistency, e.g., in studies of atmospheric temperature trends (*Ladstädter et al.*, 2011; *Khaykin et al.*, 2017; *Leroy et al.*, 2018), in climate model evaluation studies (*Lackner et al.*, 2011; *Ao et al.*, 2015; *Schmidt et al.*, 2016), and in studies of atmospheric structure and dynamics (*Scherllin-Pirscher et al.*, 2012, 2014; *Rieck et al.*,
2014; *Wilhelmsen et al.*, 2018).

     In this paper we present results from an evaluation of the ROM SAF gridded monthly-mean climate data records (CDRs), with a focus on the temporal stability of the data series and on the differences between the RO missions. The methods used to generate the gridded monthly mean data and the de-seasonalized anomalies are described, including the sampling-error correction method. The observational RO data time series are compared to the ERA-Interim reanalysis. The consistency of cli-
matologies obtained from different RO missions during mission overlap periods are studied, with a view to identify systematic differences that may have an impact on data series constructed from multiple RO missions. We also evaluate the sampling-error correction, which is essential for combining data from RO missions with different sampling characteristics.

     Section 2 provides an overview of the data that are being evaluated, and of the data used as a reference for the evaluation. In Section 3, the processing of the data to gridded monthly mean climatologies is described, including a discussion on the
time evolution of bending-angle quality and on the quality screening of the data. Section 4 describes a comparison with ERA-





**Table 1.** Low-level input data to ROM SAF CDR v1.0.

| Mission | Data provider | Version | Time period |
|---------|---------------|---------|-------------|
| CHAMP | UCAR | 2014.0140 | 2001-09 to 2008-09 |
| GRACE | UCAR | 2010.2640 | 2007-03 to 2014-03 |
| | | 2014.2760 | 2014-04 to 2016-12 |
| COSMIC | UCAR | 2013.3520 | 2006-07 to 2014-04 |
| | | 2014.2860 | 2014-05 to 2016-12 |
| Metop | EUMETSAT | 1.4 | 2006-12 to 2016-12 |
| Metop[1] | UCAR | 2016.0120 | 2008-03 to 2016-12 |

[1]Not a formal part of CDR v1.0.

Interim reanalysis data, while in Section 5 the consistency of climatologies obtained from different RO missions are analyzed. The study results are discussed and the main conclusions are summarized in Section 6.

## 2  Data

### 2.1  GPS radio occultation measurements

The ROM SAF CDR v1.0 includes data from four RO missions: CHAMP, GRACE, COSMIC, and Metop. The processing of data from the first three missions were based on low-level input data from UCAR, while the Metop data were processed with input data from EUMETSAT. In addition, we also processed Metop data using input data from UCAR. The low-level input data consist of amplitude and excess phase data, together with positions and velocities for Global Positioning System (GPS) and low-Earth orbit (LEO) satellites. The input data versions are shown in Table 1.

The input data were processed to geophysical data using the ROM SAF processing system GPAC v2.3.0, with the Radio Occultation Processing Package (ROPP) v8.1 as an integral part. The variables discussed in the present article are bending angle, refractivity, and dry temperature (the concept of "dry" variables is described in Section 3.1). The ROM SAF CDRs also contains dry pressure and dry geopotential height (the geopotential heights of dry-pressure surfaces) as well as temperature and humidity in atmospheric regions where humidity has a significant influence on the refractivity. The CDRs also include

tropopause height derived from the dry-temperature profiles, as well as from bending angle and refractivity profiles.

In total, the four RO missions include nearly 12 million occultations collected from September 2001 to December 2016. Fig. 1 shows that the mean daily number of occultations peaked at well above 3000 during 2007–2009. The launch of the second Metop satellite, in combination with an update of the operational mode of the COSMIC mission, led to a second peak in the daily data numbers in 2013. After removal of data based on quality screening, about 10 million atmospheric profiles

remain for generation of the CDRs.





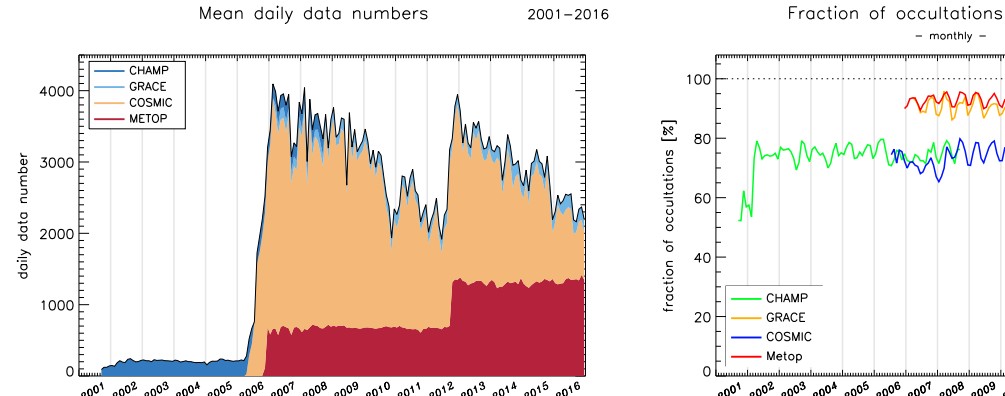

**Figure 1.** Left panel: Mean daily number of occultation events used in the generation of the ROM SAF climate data record, based on the four RO missions CHAMP , GRACE, COSMIC, and Metop. Right panel: Fraction of occultations available for the generation of gridded monthly mean data, after quality screening of the input profile data.

## 2.2 ERA-Interim reanalysis data

We used ERA-Interim reanalysis (*Dee et al.*, 2011) short-term forecasts as a reference in the evaluation. For each RO event, a co-located vertical profile of model data was obtained by interpolation in the global forecast fields representing the atmospheric state at three-hour intervals (UTC 00, 03, . . . ) on a $1.0° \times 1.0°$ latitude-longitude grid. The model data profiles are forward-modelled to the set of observed geophysical variables. This is followed by monthly averaging in latitude bins using the same methods as for the observed profiles (described in Section 3.4).

## 3 ROM SAF processing of RO data

This section provides a short description of the processing of RO measurements to atmospheric profiles of bending angles and associated geophysical variables, and further on to the gridded monthly mean data. The quality of the bending angles, and the quality screening are also briefly discussed.

## 3.1 Processing to atmospheric profiles

The input data to the ROM SAF processing consist of amplitude and excess phase time series collected during the satellite occultation events, together with precise orbits for the GPS and LEO satellites. The input data were obtained from EUMETSAT (for the Metop mission) and from UCAR (for the CHAMP, GRACE, COSMIC, and Metop missions).

Bending angles at the two GPS frequencies L1 (1575.42 MHz) and L2 (1227.60 MHz) are calculated from the excess phase and amplitude data through a geometrical optics approach (*Kursinski et al.*, 1997) above 25 km, a wave optics approach (*Gorbunov and Lauritsen*, 2004) below 20 km, and a gradual transition in between. After correction for ionospheric effects





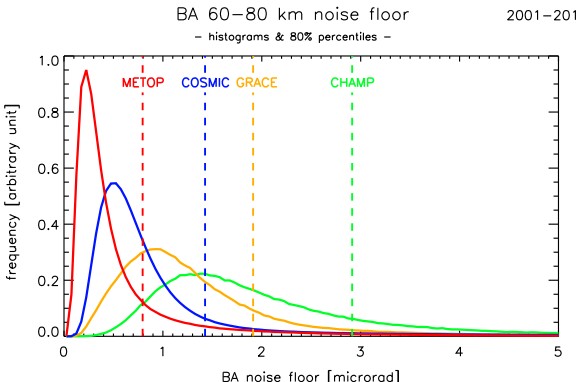

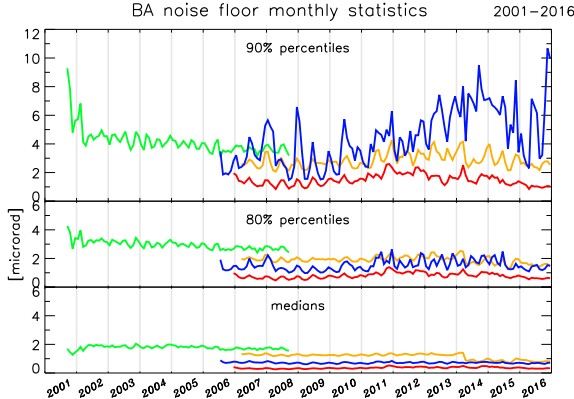

**Figure 2.** Bending angle (BA) noise floor for the RO missions CHAMP, GRACE, COSMIC, and Metop: overall distributions in the left panel, and the time evolution of the associated medians, 80% and 90% percentiles in the panel to the right. The statistics include all data, also those that are rejected in the quality screening.

through a linear combination of the L1 and L2 bending angles, we obtained the so called "raw" ionospheric corrected bending angle. With optimal linear combination (*Gorbunov*, 2002), using a bending angle climatology (*Scherllin-Pirscher et al.*, 2015), we obtain statistically optimized bending angle profiles that can be used to further retrieve geophysical information. Under the assumption of local spherical symmetry in the vicinity of the occultation point, we use the Abel transform (*Fjeldbo et al.*, 1971)

to compute a vertical refractivity profile from the bending angles. Details of the processing steps can be found in algorithm technical baseline documents (ATBDs) at the ROM SAF web site (*http://www.romsaf.org/product_documents.php*).

For a dry atmosphere the refractivity is directly proportional to air density. Dry pressure is retrieved from refractivity under the assumption of hydrostatic equilibrium and by ignoring the presence of water vapour. The corresponding dry temperatures are obtained by applying the ideal gas law. The "dry" approximation is a valid assumption in the upper troposphere and

stratosphere, where the dry variables are accurate approximations for the corresponding physical variables (*Danzer et al.*, 2014).

Under moister conditions, the dry-wet ambiguity can be resolved by a one-dimensional variational (1D-Var) retrieval (*Healy and Eyre*, 2000), using additional information from co-located ERA-Interim short-term forecasts. This gives estimates of the "wet" (physical) temperature and humidity, appropriate for atmospheric regions where humidity has a significant influence on

the refractivity.

## 3.2 Bending angle quality

The quality of the retrieved bending angles differ between RO missions. In addition to the effects of residual ionospheric noise, the quality depends on RO instrument characteristics as well as on the data processing; the use of single- or double-differencing of excess phases (e.g., *Schreiner et al., 2010*, *von Engeln et al., 2011*), and filtering of the data applied at different steps in the

processing (*Schreiner et al.*, 2011). The bending angle noise between 60 and 80 km, an altitude range where bending due to



the neutral atmosphere is small, provides and indication of the bending angle quality (*Schreiner et al.*, 2011; *Li et al.*, 2015; *Angerer et al.*, 2017). For each occultation, we compute the standard deviation of the bending angle difference with respect to a fitted background. The smallest standard deviation over any 7.5 km interval between 60 and 80 km is referred to as the bending angle noise floor for an occultation.

Fig. 2 shows the noise floor distributions for the four RO missions, and the time evolution of the associated medians and percentiles. The lower panels on the right show that 50% (80%) of the bending angle profiles have noise floors smaller than about 1.8 (2.9) µrad for CHAMP, 1.2 (1.9) µrad for GRACE, 0.7 (1.4) µrad for COSMIC, and 0.4 (0.8) µrad for Metop. These numbers are somewhat smaller than those found by *Schreiner et al.* (2011) and *Angerer et al.* (2017) due to their use of standard deviations computed over the whole 60-80 km and 65-80 km height intervals, respectively.

The bulk of the noise floor distributions are relatively constant, as shown by the time evolution of the medians. However, as indicated by the 80% and 90% percentiles time series in Fig. 2, the number of high-noise profiles is more variable. CHAMP exhibits increased noise levels during the first months of the data record. Following an instrument software update in March 2002, the bending angle noise settles at an almost constant level, but with the number of high-noise profiles slowly declining. The bending angle noise in the GRACE data is relatively constant, except for a sudden decrease in April 2014 which affects the

bulk of the profiles, not only the number of noisy profiles. This stepwise change is due to a switch to zero-differencing in the generation of the excess phases in UCAR's version 2014.2760 of GRACE data. For COSMIC, there is a substantial increase with time of a the number of profiles with very high bending angle noise, mainly attributed to rising occultations (not shown here). Metop exhibits a somewhat larger number of high-noise profiles for rising than for setting occultations, and also shows an interesting pattern which may be related to the solar cycle. A similar pattern may be discernible in the GRACE data.

## 20 3.3 Data quality screening

Before processing the atmospheric profiles to gridded monthly-mean data, all profiles are checked against a set of quality criteria. The quality criteria include tests to identify occultations that a) are obviously corrupt or show signs of major problems, b) have degraded bending angles, c) could be regarded as outliers, or d) encounter problems in the 1D-Var processing. More detailed descriptions of the data quality screening are found in the series of validation reports available at the ROM SAF web

site (*http://www.romsaf.org/product_documents.php*).

  If an occultation does not pass one or several of the tests in a, b, or c, the bending angle, refractivity, and dry variables are marked as non-nominal. Otherwise, they are regarded as nominal and the refractivity profiles are passed on to the 1D-Var processing, followed by the associated quality tests. The fraction of data rejected in the quality screening varies over time and between the RO satellite missions (Fig. 1).

## 30 3.4 Monthly averaging in latitude bins

The gridded monthly mean data are obtained by a simple binning-and-averaging technique. Each occultation is assigned to a 5-degree latitude band and calendar month. The RO profiles that pass the quality screening are interpolated onto an equidistant 200 meter height grid (the height variable is impact altitude for bending angles, pressure height for dry geopotential height, and




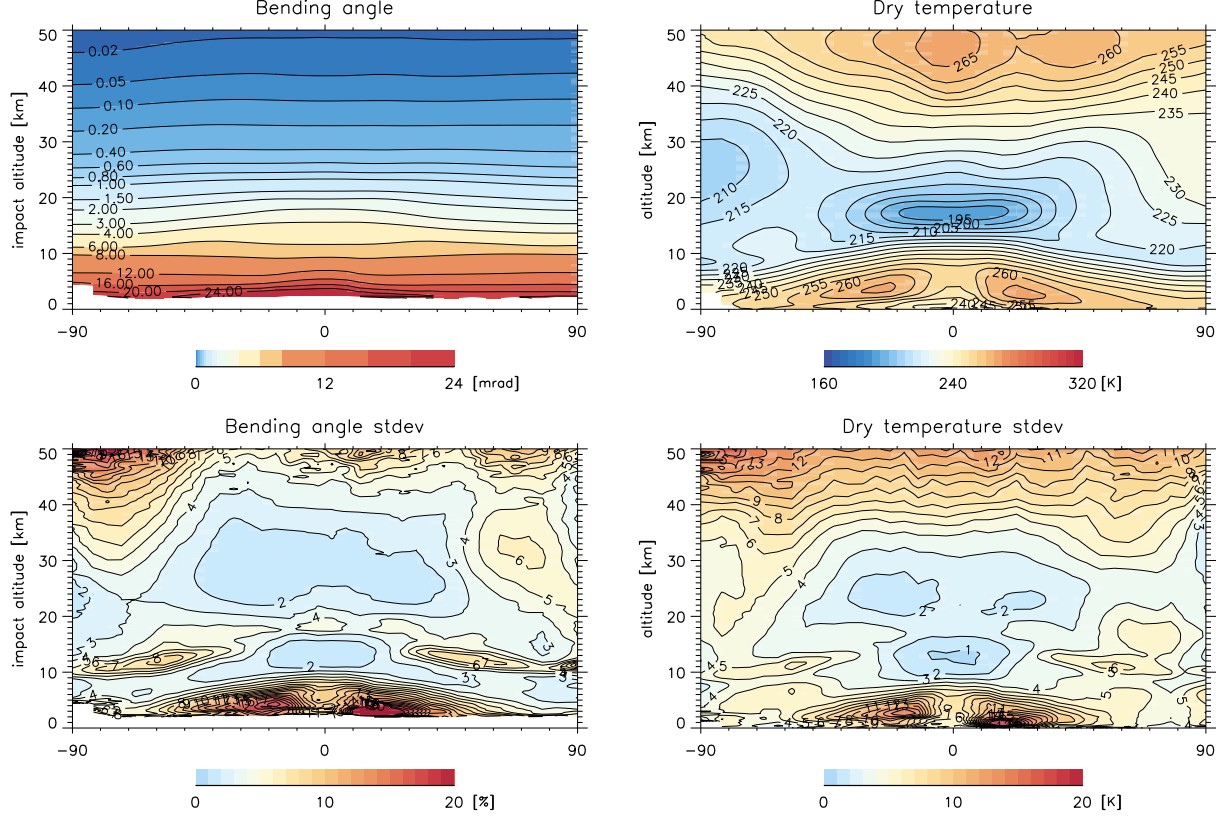

**Figure 3.** Zonally gridded monthly means and standard deviations for bending angle (left panels) and dry temperature (right panels) in April 2014. The monthly means were sampling-error corrected, and include data from the Metop-A and Metop-B satellites.

mean-sea level altitude for the other geophysical variables). At each height, and within each bin, the data undergo a weighted averaging. The purpose of the weighting is to reduce the effects of a non-uniform spatial sampling density across a grid box, in order to better approximate an area-weighted mean. The distribution of observations in longitude is nearly uniform and is not explicitly addressed. The distribution of observations in latitude, on the other hand, can be highly non-uniform. This is

5 addressed by sub-dividing each 5-degree latitude bin into two sub-bins, and giving each data point, $i$, a weight, $w_i$, according to which sub-bin, $s$, it belongs to:

$$w_i = \frac{A_s}{A} \frac{n}{n_s} \qquad (1)$$

where $A$ and $n$ are the total area and data number for the bin, and $A_s$ and $n_s$ are area and data number for sub-bin $s$. Within each latitude bin and calendar month, a weighted arithmetic average is computed as

10 $$\bar{X}(h) = \frac{\sum w_i X_i(h)}{\sum w_i} \qquad (2)$$





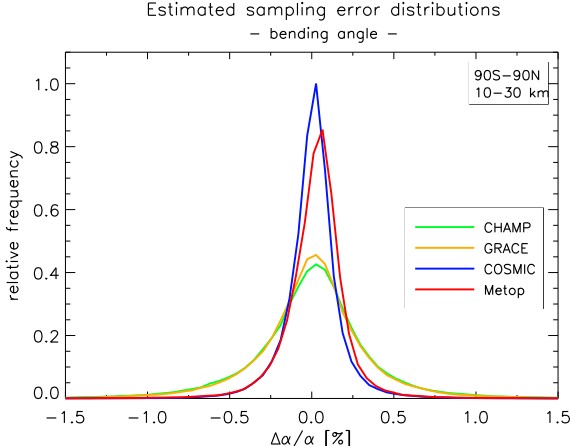
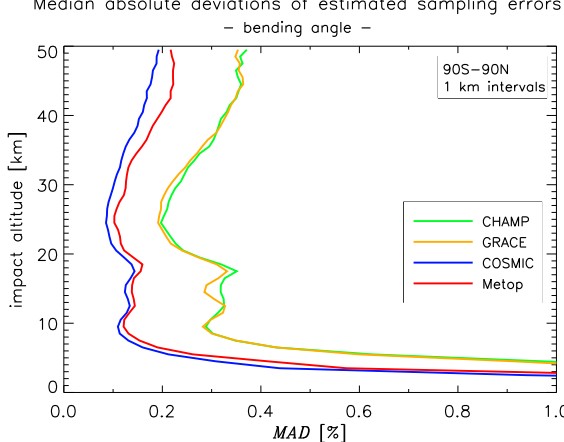

**Figure 4.** The panel to the left shows distributions of estimated bending angle sampling errors based on all monthly bins in the 10-30 km height interval, for the four RO missions CHAMP, GRACE, COSMIC, and Metop. The spread of the distributions, as quantified by the median absolute deviation (MAD), is about 0.3% for CHAMP and GRACE and 0.1% for COSMIC and Metop. The right-hand panel shows the MADs for height resolved sampling-error distributions.

where $X_i$ is a geophysical quantity, and $\bar{X}$ is the corresponding monthly mean for the latitude bin. The corresponding weighted standard deviation is given by

$$s(h) = \sqrt{\frac{\sum w_i (X_i(h) - \bar{X}(h))^2}{((n-1)/n)\sum w_i}} \tag{3}$$

using the same weights as in Eq. 2. The dependency of the weights, $w_i$, and the data numbers, $n$, on height is not shown explicitly in the above equations. Fig. 3 shows an example of bending angle and dry-temperature means and standard deviations for Metop data from April 2014.

### 3.5 Sampling errors and sampling-error correction

The finite number of observations is not enough to fully account for all variability within the time-latitude bins, leading to a sampling error in the monthly means. The sampling error, $\varepsilon_{\mathrm{samp}}$, can be estimated by sampling a model atmosphere at the same times and locations as the observations, and then subtract the true model monthly mean from the monthly mean based on the sampled model data,

$$\varepsilon_{\mathrm{samp}} = \bar{X}_{\mathrm{model}}^{\mathrm{samp}} - \bar{X}_{\mathrm{model}}^{\mathrm{true}} \tag{4}$$

The sampled monthly mean in Eq. 4 is constructed similarly to the observed monthly mean, using the methods described in Section 3.4. The true model mean for a monthly bin is computed from the full 4-dimensional reanalysis model field,

$$\bar{X}_{\mathrm{model}}^{\mathrm{true}} = \frac{1}{n_t n_\varphi n_\lambda} \cdot \frac{1}{\sum_{k=1}^{n_\varphi} \cos\varphi_k} \cdot \sum_{t=1}^{n_t} \sum_{k=1}^{n_\varphi} \sum_{l=1}^{n_\lambda} X_{tkl} \cos\varphi_k \tag{5}$$





where $\varphi_k$ is the latitude at a model grid point, and the summation loops over all model grid points located within the 5-degree latitude band for that calendar month. Similar techniques for sampling-error estimation have been described by, e.g., *Foelsche et al.* (2003, 2008), *Scherllin-Pirscher et al.* (2011), and *Ho et al.* (2009). In the ROM SAF CDR, sampling errors are estimated from ECMWF reanalysis short-term forecast fields (currently, ERA-Interim) at a $2.5° \times 2.5°$ latitude-longitude grid and a 6 hour time step.

The accuracy of the estimated sampling error, $\varepsilon_{\mathrm{samp}}$, depends on the ability of the model to describe the true atmospheric variability within the monthly bins, at the spatio-temporal resolution of the observations. This includes both synoptic-scale variability as well as various modes of cyclic variability, e.g., the atmospheric diurnal and semi-diurnal cycles. The accuracy of the mean state within the bins is less important as it is largely removed by the subtraction in Eq. 4.

The magnitude of the sampling error depend on how well the dominating modes of variability are sampled. The errors are smaller for COSMIC than for CHAMP or GRACE, mainly due to larger data numbers, but also due to a better sampling of the diurnal cycle. The left-hand panel of Fig. 4 shows the distributions of estimated sampling errors for the four RO missions, computed from all monthly bins within a core region of 10-30 km. The bending angle spread, as measured by the median absolute deviation (MAD), is about 0.3% for CHAMP and GRACE and 0.1% for COSMIC and Metop. The corresponding numbers for dry temperature are 0.3 K and 0.1 K, respectively (not shown here). However, a small number of bins have much larger sampling errors, mainly associated with wintertime mid- and high-latitude variability. The panel to the right in Fig. 4 shows the deviations for height resolved sampling error distributions. The estimated sampling errors are smallest in a region between 10 and 35 km.

The estimation of sampling errors by means of a model provides an opportunity to do a partial correction of this important class of error. Such a correction, or adjustment, can potentially reduce systematic biases between climatologies obtained from different RO missions with different sampling characteristics and reduce systematic bias changes as the global RO constellation changes with time. Sampling-error corrected means are computed by subtracting the estimated sampling error from the observed mean

$$\bar{X}^{\mathrm{corr}} = \bar{X} - \varepsilon_{\mathrm{samp}} \tag{6}$$

The consequence of the correction is clearly seen when comparing gridded monthly means computed from disjoint sets of RO observations, e.g., monthly means computed from different RO missions during overlap periods. This is further discussed in Section 5.3 where it is shown that sampling-error correction significantly decreases inter-mission differences, leaving a residual sampling error, $\varepsilon_{\mathrm{resamp}}$, that may be handled as a quasi-random, statistical error.

### 3.6 Anomaly data time series

The gridded monthly mean RO data records discussed in this paper can be described as time series of variables on a two-dimensional latitude-height grid

$$\bar{X}_{ijm} = f(\varphi_i, h_j, m) \tag{7}$$





where $\bar{X}_{ijm}$ is a monthly-mean climate variable (e.g., refractivity or dry temperature). Indices $i$ and $j$ denote the latitude and height bins (with reference latitude, $\varphi_i$, and height, $h_j$, respectively) and $m$ denotes the time (a running month number). The anomalies are defined as the deviations from a climatological seasonal cycle, $\bar{X}_{ijs}^{\mathrm{clim}}$, where $s = 1, \ldots, 12$ is the season (month of the year). Hence, the anomalies are given by

$$\bar{X}_{ijm}^{\mathrm{anom}} = \bar{X}_{ijm} - \bar{X}_{ijs}^{\mathrm{clim}} \tag{8}$$

and the fractional anomalies are given by

$$\bar{X}_{ijm}^{\mathrm{anom}} = (\bar{X}_{ijm} - \bar{X}_{ijs}^{\mathrm{clim}})/\bar{X}_{ijs}^{\mathrm{clim}} \tag{9}$$

where the latter is the preferred expression for variables that have a dominating exponential altitude dependence.

The mean seasonal cycle, i.e. the long-term mean state as a function of latitude, height, and season, is constructed from RO data itself, although it may be based on a different combination of RO missions:

$$\bar{X}_{ijs}^{\mathrm{clim}} = \frac{1}{N_{\mathrm{yr}}} \sum_{k=1}^{N_{\mathrm{yr}}} \bar{X}'_{ijm} \quad , \quad m = 12 \cdot (k - 1) + s \tag{10}$$

where $N_{\mathrm{yr}}$ is the number of years used in the generation of the climatology. In the generation of anomaly time series, the same seasonal cycle should be used for all missions and throughout the time series. This is particularly important for investigations of differences between RO missions.

The anomalies depend on latitude, altitude, and time. Averaging over a latitude band, properly weighting the latitude bins proportional to their areas, gives a two-dimensional time-altitude data set (see Fig. 7), while averaging over both a latitude band and a height layer gives a one-dimensional time series (see Figs. 9 and 10).





**Figure 5.** Globally averaged monthly mean bending angle (left panels) and refractivity (right panels) differences with respect to ERA-Interim, for the four RO missions included in the ROM SAF CDR. From bottom to top, the panels show vertically averaged differences in the 4-8 km, 8-12 km, 12-20 km, 20-30 km, 30-40 km, and 40-50 km intervals, where the height coordinate is impact altitude for bending angle and altitude for refractivity. Note that for the lowest bending angle panel (left column), the vertical axis has been compressed to accommodate the larger differences between the missions.

## 4    Comparison with ERA Interim reanalyses

The ROM SAF CDR is evaluated using the ERA-Interim reanalysis as a reference, with the purpose to provide a better understanding of the time evolution of the RO data and the stability in time. As a side-effect, time-varying biases and sudden bias

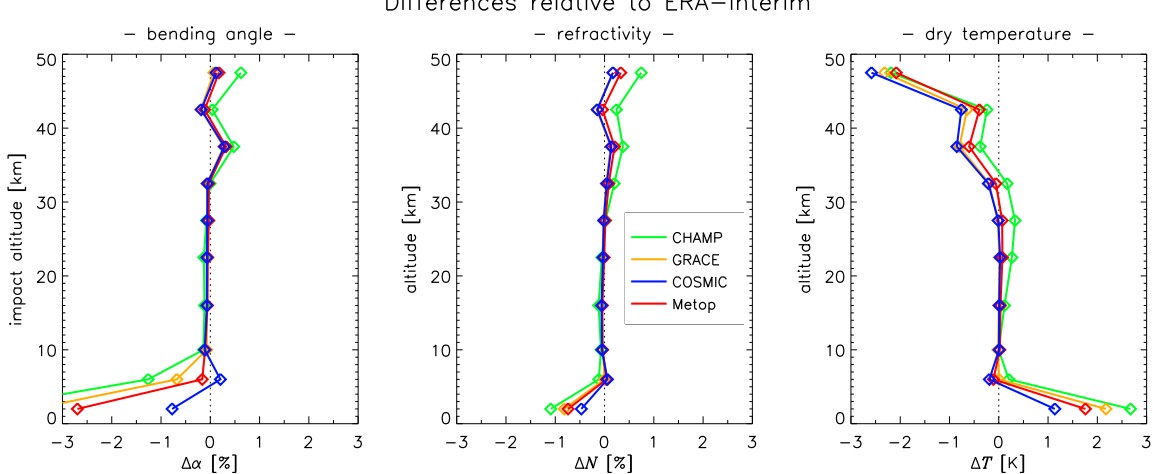

**Figure 6.** Globally averaged monthly mean bending angle (left panel), refractivity (middle panel), and dry temperature (right panel) differences relative to ERA-Interim, for the four RO missions included in the ROM SAF CDR. The profiles show time averages, with the averaging period chosen as the full length of the respective satellite mission.

shifts in the ERA-Interim reanalysis data are identified. In addition, the comparison of all four RO missions against the same reference provides an indication of the consistency of RO climatologies generated from different missions.

Fig. 5 shows globally averaged relative differences of observed bending angle and refractivity with respect to ERA-Interim, for the four RO missions and for six altitude layers from 4 to 50 kilometers. The smallest differences are found at altitudes between about 8 and 30 km. In this altitude region, the dominating features in the difference time series reflect bias shifts in the ERA-Interim data itself. In December 2006 the magnitude of the bias suddenly decreased in the 12-20 km and 30-40 km intervals. In November 2009 there was a shift from a small negative bias to a near zero difference in the 20-30 km interval. Then in late 2013 the difference suddenly dropped back to a small negative bias level during a few weeks. These events are most likely related to the start of assimilation of COSMIC data in 2006 (*Dee et al.* (2011)), an update of the COSMIC NRT data processing in October 2009 (*Healy* (2013)), and a temporary drop-out of RO data from the ERA-Interim assimilation system in late 2013 (*S. Healy, pers. comm.*). Above 30 km, the differences between RO and ERA-Interim are larger, particularly for the earlier pre-COSMIC time period, when the impact of RO on the reanalysis was weaker due to lower data numbers.

In the 8 to 40 km altitude interval, the spread amongst the RO missions is generally smaller than the differences between RO and ERA-Interim. In combination with the fact that the dominating shifts in the difference time series can be attributed to ERA-Interim, this suggests that the RO data have better long-term stability than the ERA-Interim data. At the highest altitudes, above 40 km, the larger differences in the earlier time period and the smaller differences later on, lead to a long-term trend in the differences. Below 8 km, the spread amongst the RO missions are larger, with the COSMIC and Metop data showing a better match to ERA-Interim than the CHAMP and GRACE data. The CHAMP data series exhibits bias shifts in March 2002



and in July 2006. The former shift coincides with a firmware update of the GPS-RO instrument onboard the CHAMP satellite (*J. Wickert, pers. comm.*).

Fig. 6 shows the RO versus ERA-Interim differences in the form of global time-averaged vertical profiles for bending angle, refractivity, and dry temperature. The profiles for the different missions do not represent identical time periods, as the time averaging is done over the full length of the respective satellite mission. In line with the findings in Fig. 5, the CHAMP profiles are somewhat deviating from the profiles of the other satellite missions, particularly above 35 km for bending angle and refractivity and also at lower altitudes for dry temperature. Of the four RO missions, CHAMP also exhibits the largest lower-tropospheric biases relative ERA-Interim.

Regarding the stability in time, it should be noted that even though the ERA-Interim reanalysis system in itself does not change with time, the evolving global observing system lead to time-varying biases (*Dee et al.*, 2011). ERA-Interim does not provide a stable enough reference against which to accurately measure temporal stability of the RO data. Between about 8 and 30 km, the RO data records are likely to have a higher temporal stability than ERA-Interim. At higher and lower altitudes, the long-term temporal stability of the multi-mission RO time series is limited by the evolving global RO constellation, and depends on the magnitude and character of the differences between the RO satellite missions. This is discussed Section 5.

## 5 Differences between RO missions

Differences in the monthly means obtained from RO missions that overlap in time are due to a combination of random profile errors, sampling errors, and systematic errors of instrumental or data-processing origin. While random errors contribute to a general degradation of the quality of the climatologies, they do not prevent us from combining data from different missions. Systematic errors on the other hand can, potentially, introduce time-evolving biases in combined multi-mission data records. In this section some of the RO mission differences, detected from mission overlaps, are identified. The influence of these differences on time series of bending angle and dry-temperature anomalies is assessed. The sampling-error correction method, described in Section 3.5, is also evaluated and its efficiency in reducing differences between the RO missions is investigated.

### 5.1 Time-altitude bending angle plots

Fig. 7 shows global monthly mean bending angle anomalies for the four RO missions. The four data records are structurally very similar above about 6-8 km and below about 35-40 km impact altitude. This altitude span encompasses the core region from the middle troposphere to the lower/middle stratosphere where RO measurements are known to have the highest quality (e.g., *Kuo et al.* (2005), *Scherllin-Pirscher et al.* (2011)). In the lowest few kilometers, there are known biases in bending angle observations obtained in moist low-latitude regions, leading to substantial differences between the missions. Throughout the stratosphere, Metop and COSMIC are qualitatively very similar, while GRACE and CHAMP exhibit quasi-random, noise-like, structures above about 35 km.

Fig. 8 shows bending angle differences between GRACE and COSMIC (left column), between Metop and COSMIC with Metop based on input data from UCAR (middle column), and between Metop and COSMIC with Metop based on input data

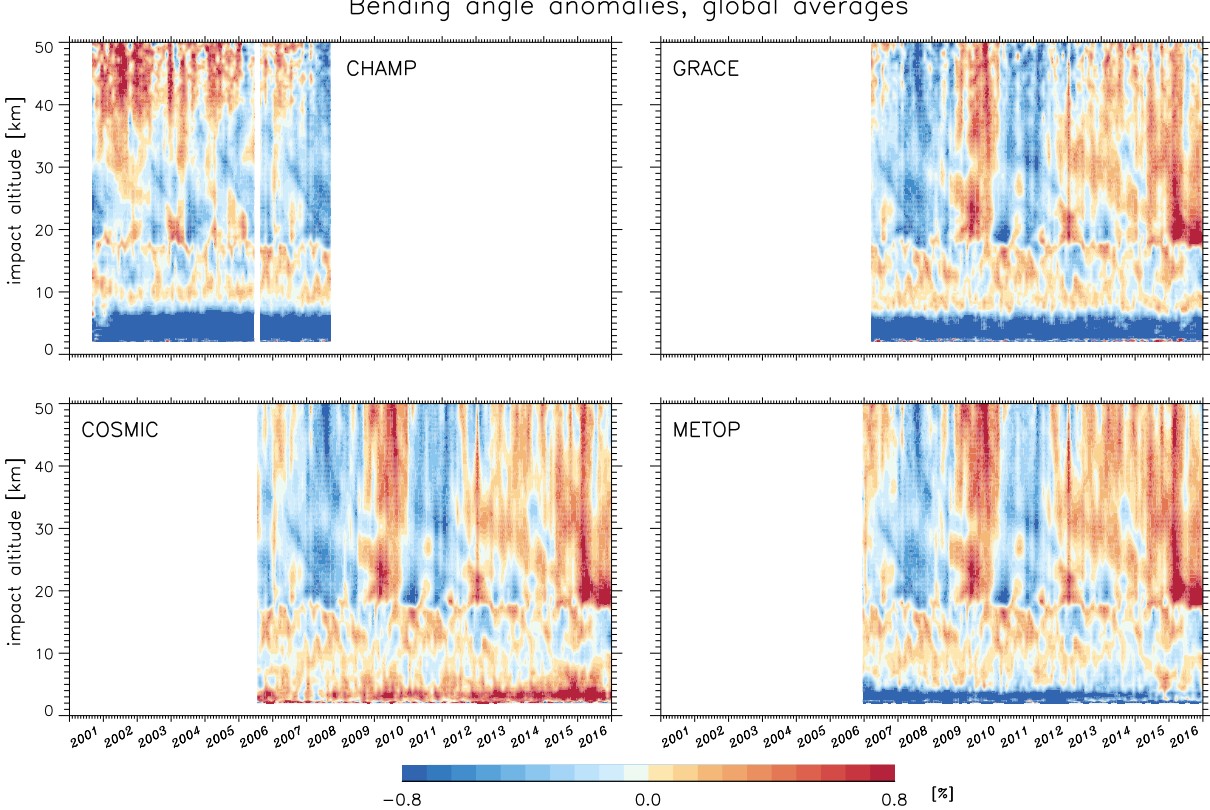

**Figure 7.** Global monthly mean bending angle anomalies, for the four RO missions included in the ROM SAF Climate Data Record v1.0. There is overlap between CHAMP and COSMIC from August 2006 to September 2008, between GRACE and COSMIC from March 2007 to December 2016, and between Metop and COSMIC from December 2006 to December 2016. Data are given on a 5 degrees by 200 meter latitude-altitude grid and have been sampling-error corrected. The anomalies for the four satellite missions are computed based on the same reference climatology, obtained from RO data itself.

from EUMETSAT (right column). COSMIC is chosen as comparison reference because it provides the longest record of the four missions, and because it has a good local-time coverage. Above 6-8 km impact altitude, the differences between the RO missions are small (note that the bending angle color range in the plots only spans $\pm 0.2\%$). A large fraction of the variability in the difference plots consists of a quasi-random, noise-like pattern, with a broad minimum between 10 and 25 km altitude. This

5   pattern is most evident in the GRACE-COSMIC plots (Fig. 8, left column). The quasi-random pattern is also present in the Metop-COSMIC plots (Fig. 8, middle and right columns), but is less visible as it is superposed on an almost uniform positive bias level (red colors) at low- and mid-latitudes.

The difference plots in Fig. 8 reveal a range of systematic differences between RO missions that cannot be explained by random profile errors or by quasi-random sampling effects. We identify the following systematic bending angle biases:



**Figure 8.** Differences between monthly-mean bending angle climatologies retrieved from different missions; between GRACE and COS-MIC (left column), between Metop(ucar) and COSMIC (middle column), and between Metop(eum) and COSMIC (right column), where Metop(ucar) is processed by ROM SAF based on input data from UCAR and Metop(eum) is based on input data from EUMETSAT. Differences are shown in five latitude bands, south to north from lowest panel to top panel.

- Biases in the lower troposphere up to a few percent (out of scale). The biases are stronger and have a larger vertical extent at low latitudes, and are believed to be linked to signal tracking issues in moist regions of the atmosphere.

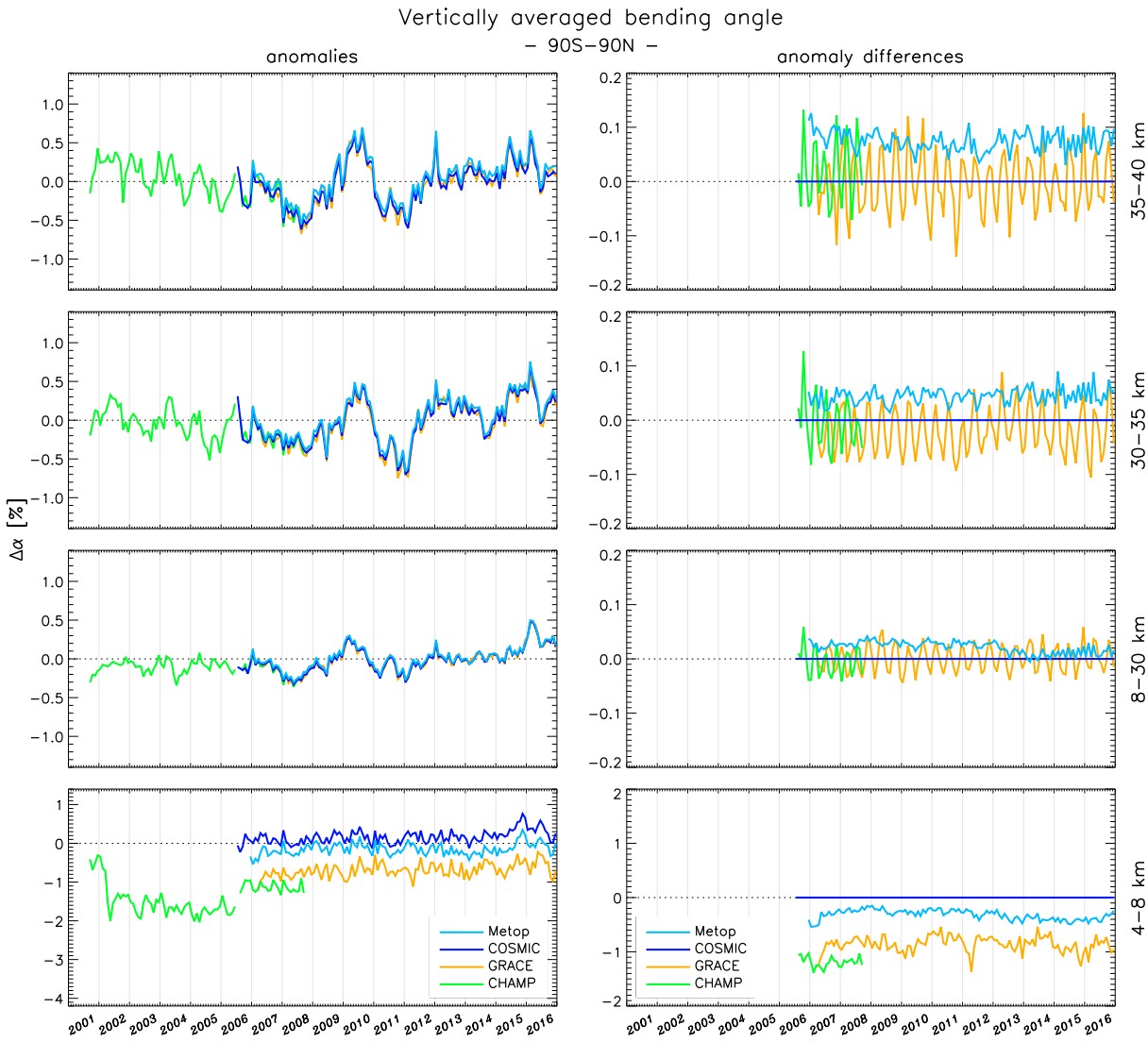

**Figure 9.** Monthly mean bending angle anomalies for the four RO missions (left panels) and differences of CHAMP, GRACE, and Metop with respect to COSMIC (right panels). From bottom to top, the panels show the 4-8 km, 8-30 km, 30-35 km, and 35-40 km height layers. Note that for the lowest plots, the vertical axes have been compressed to accommodate the larger differences between the missions.

- Seasonally varying biases up to 0.1-0.2% at high altitudes (>30 km) and high latitudes (>60 degrees latitude).

- Bias shifts on the order of 0.1% below about 20 km in the Metop-COSMIC differences in 2013. These shifts are related to firmware upgrades of the RO instruments onboard Metop-B (in April 2013) and Metop-A (July 2013). The tracking

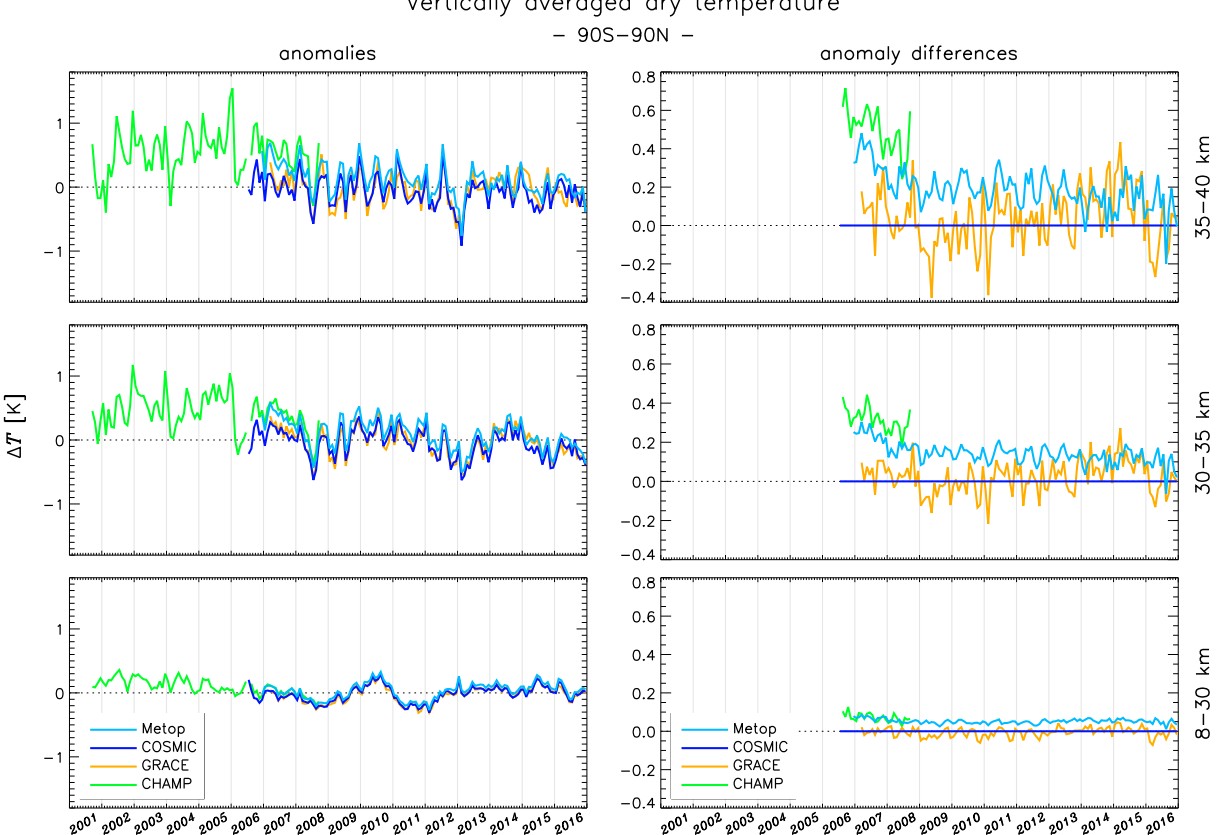

**Figure 10.** Monthly mean dry temperature anomalies for the four RO missions (left panels) and differences of CHAMP, GRACE, and Metop with respect to COSMIC (right panels). From bottom to top, the panels show the 8-30 km, 30-35 km, and 35-40 km height layers.

of the GPS signals for the Metop rising occultations was changed, which had the effect that a minor bias due to the L2 extrapolation in the ROM SAF processing suddenly appeared.

- Large-scale hemispherically asymmetric (north-south) Metop-COSMIC bias on the order of 0.1% above 35-40 km, and increasing upward. Only seen in the plots with Metop data based on input from EUMETSAT (Fig. 8, rightmost column). This difference is believed to be related to differences in LEO satellite orbits from the two sources of input data.

- Relatively uniform bias in Metop-COSMIC differences at low- and mid-latitudes on the order of 0.03% at 20 km and increasing upward (0.1% at 40 km). Believed to be related to under-sampling of the diurnal cycle, in combination with imperfect sampling-error correction of the Metop data.

- GRACE-COSMIC and CHAMP-COSMIC cyclic differences (the latter not shown here) at low- and mid-latitudes on the order of 0.03% at 20 km and increasing upward. This is a weak effect and is just barely seen in Fig. 8 (vertical



averaging makes this effect more easily detected, see Fig. 9). The cycle period is around 5 months for GRACE-COSMIC and 4 months for CHAMP-COSMIC. Believed to be related to under-sampling of the diurnal cycle, in combination with imperfect sampling-error correction.

Most of these RO mission differences are caused by systematic errors in the underlying profile data, that are propagated to

the gridded monthly means. The exception is the sampling errors that are intrinsic to the gridded data.

## 5.2 Anomaly time series

Sofar, the RO mission differences have been described in terms of bending angles. However, the identified differences are also relevant for the geophysical variables retrieved from bending angle, e.g., refractivity and temperature. Generally, errors in bending angle propagate downward to lower altitudes in the retrieval chain. This becomes evident in the anomaly time series

discussed below.

### 5.2.1 Bending angle

The left column of Fig. 9 shows globally averaged monthly-mean anomalies of bending angle, vertically averaged in four height layers: 4-8 km, 8-30 km, 30-35 km, and 35-40 km. Each plot includes data for the four RO missions. The corresponding panels in the left column of Fig. 9 show the differences of CHAMP, GRACE, and Metop with respect to COSMIC. During the

years 2008 to 2014, the six-satellite COSMIC constellation also had a nearly complete local-time coverage, which is important for detecting the impact of under-sampling the diurnal cycle in the other missions.

In 8-30 km vertically averaged data (Fig. 9, third row from top), the four time series show a very close match. How well the overlapping time series match must be evaluated in relation to the variability of the time series itself. There is variability on a broad range of time scales, from short-range intra-seasonal variations to inter-annual and decadal variability, and long-term

climatological trends. We find that for the 8-30 km time series, the mean (time averaged) differences between the missions are -0.005%, 0.001%, and 0.02%, respectively, for CHAMP, GRACE, and Metop relative to COSMIC. This is much smaller than the intrinsic variability of the time series – the total range of global monthly mean bending angle anomalies in this height range is about 1%. Metop shows a relatively steady bias relative to COSMIC, with a stepwise decrease of the bias in mid-2013, and with a tendency to oscillations after 2014. The CHAMP and GRACE differences with respect to COSMIC exhibit strong

oscillating behaviour with peaks that reach about the same magnitude as the Metop-COSMIC bias. The cycle periods for the oscillating difference time series are about 4 months and 5 months, respectively, closely corresponding to the precession rate of the respective satellite orbit. This could be explained as a consequence of the sampling-error correction not being able to fully compensate for the effects of under-sampling the diurnal and semi-diurnal cycles. It would also be consistent with the near-constant Metop-COSMIC biases because of the Sun-synchronous Metop orbit.

Above the middle troposphere, the mean differences in Fig. 9 increase with altitude, and for Metop-COSMIC the differences are about 0.04% in the 30-35 km interval and 0.08% in the 35-40 km interval. The mean GRACE-COSMIC differences are small, while the mean CHAMP-COSMIC differences increase to 0.02% in the 35-40 km interval. The magnitude of the



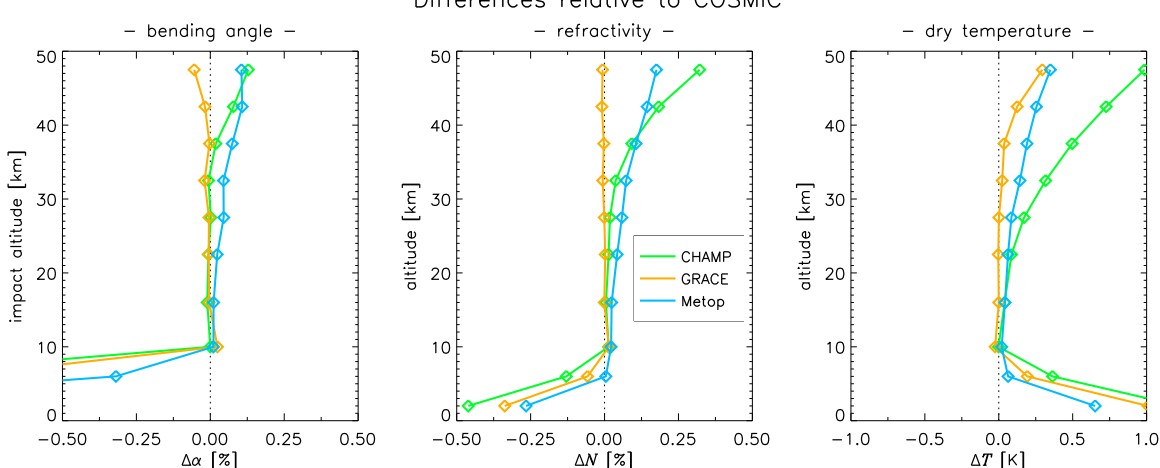

**Figure 11.** Summary of global CHAMP, GRACE, and Metop differences relative to COSMIC, quantified by the time averages of the mission differences computed over the respective overlap period. The RO data have been binned, averaged, sampling-error corrected, and converted to anomalies according to the methods described in Section 3.

CHAMP-COSMIC and GRACE-COSMIC oscillations increases with altitude, such that the peak biases of these two mission differences reach about the same values as the Metop-COSMIC biases.

In the tropospheric 4-8 km impact altitude interval (Fig. 9, left column), we find relatively large biases between the missions. Monthly global averages of CHAMP and GRACE bending angles are about 1% smaller than the corresponding COSMIC data, while Metop bending angles are about 0.3% smaller. It should also be noted that the CHAMP data record itself shows substantial bias shifts in 2002 and 2006.

The left panel of Fig. 11 summarizes the mission differences for global mean bending angle anomalies. The summary is based on the time-averaged differences computed over the respective overlap period. The RO mission consistency, defined as the largest time-averaged difference between any two missions, is about 0.04% above 8 km and below 30 km, increasing to 0.08% below 40 km, and about 0.18% below 50 km. These numbers can be up to a factor of two larger for 30-degree latitude bands compared to global means.

### 5.2.2 Refractivity

Similarly to the bending angle anomalies, the global refractivity anomalies for the four RO missions show a very close match in the 8-30 km vertically averaged data (not shown here). The differences of the refractivity anomalies for Metop relative to COSMIC are nearly constant (about 0.04%) over the 10 years of overlap, with just a small decrease of the differences after mid-2013, and with a tendency to short time-scale oscillations towards the end of the time period. The differences of CHAMP and GRACE relative to COSMIC show similar, steady oscillation patterns as the corresponding bending angle time series.





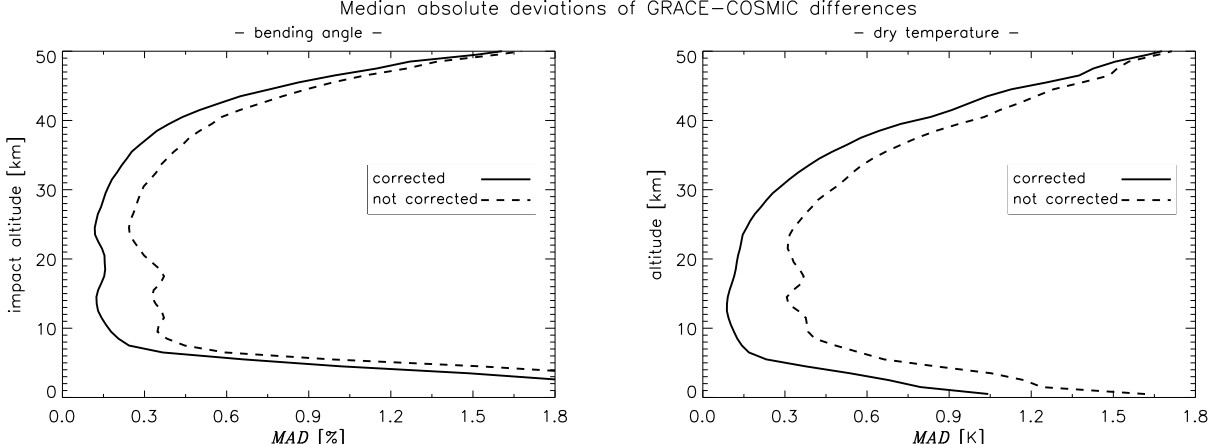

**Figure 12.** Median absolute deviations of GRACE-COSMIC differences computed from all monthly bins in 1-km height intervals during the mission overlap period March 2007 to December 2016; deviations for bending angles to the left and for dry temperature to the right. The solid lines show deviations with sampling-error correction applied, and the dashed lines show deviations without correction.

Vertically averaged global refractivity anomalies for 5-kilometer layers from 30 to 50 km show a systematic increase of the mission differences with height (not shown here). This is similar to the bending angles, although the differences are larger for refractivity due to the downward propagation of errors in the Abel transform (Section 3.1).

The middle panel of Fig. 11 summarizes the mission differences for global mean refractivity anomalies. The consistency is
5 about 0.05% between 8 km and 30 km, increasing to 0.11% below 40 km, and about 0.32% below 50 km. These numbers can be up to a factor of two larger for 30-degree latitude bands compared to global means.

### 5.2.3 Dry temperature

The globally averaged dry-temperature anomalies are shown in Fig. 10. The RO differences in the 8-30 km vertical averages are smaller than 0.10 K. We note that the oscillating behaviour seen in the bending angle and refractivity differences for
CHAMP and GRACE relative to COSMIC is there also for dry temperature, but is less obvious as it has a much more irregular appearance. At higher altitudes, the dry-temperature anomalies in 5-kilometer layers from 30 to 50 km show increasingly larger differences between the RO missions. In addition to the errors propagated from bending angle to refractivity, there is also a downward propagation of errors due to the hydrostatic integration used in the retrieval of dry temperature.

The right-hand panel of Fig. 11 summarizes the mission differences for global mean dry temperature anomalies. The RO
mission consistency is about 0.15 K between 8 km and 30 km, increasing to 0.30 K up to 40 km , and 0.50 K up to 40 km. These numbers can be up to a factor of two larger for 30-degree latitude bands compared to global means.



### 5.3 Evaluation of the sampling-error correction

Mission differences during overlap periods allow us to investigate some of the consequences of sampling-error correction, and to assess the magnitude of the residual sampling errors remaining after correction. Fig. 12 shows the median absolute deviations of GRACE-COSMIC differences based on all monthly bins in 1-km height intervals during the mission overlap period March

2007 to December 2016. The solid lines are computed from GRACE and COSMIC data with sampling-error correction applied, while the dashed lines are computed from data without correction. The application of sampling-error correction substantially reduces the GRACE-COSMIC differences, both for bending angle and dry temperature, as well as for other geophysical variables (not shown).

The deviations remaining after sampling-error correction (indicated by the solid lines in Fig. 12) are due to a combination of

GRACE and COSMIC random profile errors, residual sampling errors, and any systematic differences between the RO missions that have a sufficiently strong variation with time and/or latitude. In the core region 8-30 km, random profile errors can at most explain a part of the 0.1-0.2% deviations for the bending angles and the 0.10-0.15 K deviations for dry temperature. Assuming, conservatively, that these remaining errors are due solely to residual sampling errors, we find that around one third of the original sampling error remain after sampling error correction. This is roughly in line with the findings of *Scherllin-Pirscher*

*et al.* (2011).

Fig. 13 shows an example of the consequences of sampling-error correction for the mission differences. The left-hand panel of Fig. 13 shows mission difference anomaly time series without correction, and the right-hand panel show the same differences with the correction applied. The CHAMP and GRACE differences relative to COSMIC are dominated by a periodic oscillation, presumably due to aliasing between the LEO satellite orbital precession and diurnal or semi-diurnal cycles in the atmosphere.

The magnitude of these oscillations are substantially reduced by the sampling-error correction. However, the fact that they are not entirely removed indicates that the estimated sampling errors, based on sampling ERA-Interim reanalysis fields, are not able to fully capture the diurnal and/or semi-diurnal cycles. Unlike the case with CHAMP and GRACE, the differences between Metop and COSMIC are not periodic. Given the Sun-synchronous orbit of Metop, the errors resulting from under-sampling of the diurnal cycle are expected to be near-constant in time, which is roughly in line with the findings.

### 6 Summary and conclusions

In this study, we present results from an evaluation of the 15-year ROM SAF CDR v1.0 consisting of separate data records from four different RO satellite missions: CHAMP, GRACE, COSMIC, and Metop. The processing of the RO data to gridded monthly means is described, including the sampling-error correction of the monthly mean data. The observed bending angle, refractivity, and dry-temperature records are compared to the ERA-Interim short-term forecasts. The four RO data records are

also inter-compared during mission overlap periods and the impact of the sampling-error correction, applied to the gridded monthly mean data, is evaluated.

In general, there is good overall agreement between the ROM SAF gridded monthly mean CDR and the ERA-Interim reanalysis, particularly in the 8-30 km height interval. Here, the differences appear to mainly reflect time-varying biases in





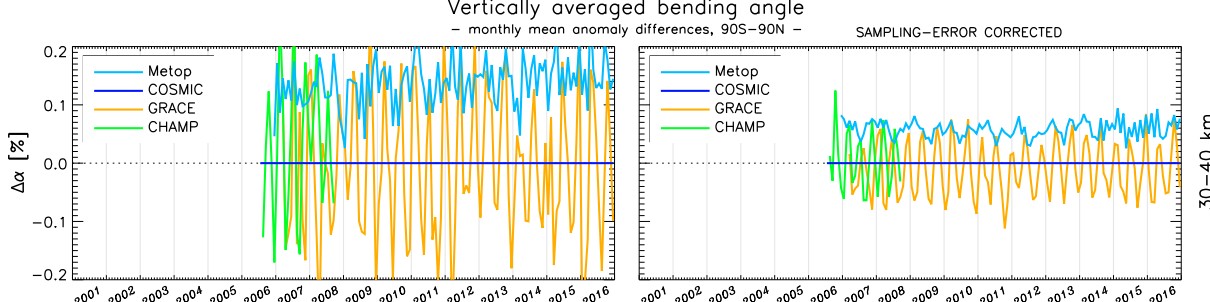

**Figure 13.** Differences between monthly mean CHAMP, GRACE, and Metop refractivity anomalies relative to COSMIC, globally averaged in the 30-40 km altitude layer. Left-hand panel without sampling-error correction and to the right with sampling-error correction applied.

ERA-Interim, as indicated from the timing of the bias shifts and the fact that the spread amongst the RO missions is smaller than the differences between RO and ERA-Interim. We interpret this as a better temporal stability in the RO data records than in the ERA-Interim time series. At high altitudes, above 30-40 km, we find larger differences between RO and reanalysis, and also a long-term trend in the difference time series. At altitudes below 8 km the differences are again larger, particularly for

bending angle, with a relatively large spread amongst the RO missions.

To fully exploit the RO data records scientifically requires that we can combine the data records from several RO missions into multi-mission data records. There is an expectation that this can be done without any adjustments or inter-calibrations. However, any differences between the missions in the retrieved geophysical data may lead to time-varying biases in the multi-mission data record as new satellite missions replace older ones. We investigated the presence of such differences during

mission overlap periods and found that there is a high degree of consistency between the RO satellite missions in the 8-30 km altitude region. The remaining differences in this altitude interval are predominantly oscillatory or highly variable for CHAMP and GRACE relative to COSMIC, while for METOP the differences relative to COSMIC largely consist of small, but stable, offsets. These differences should be considered in the generation of multi-mission data records. At higher altitudes the differences between the RO missions become increasingly larger, and at altitudes below 8 km we find biases and bias shifts

that substantially reduces the inter-mission consistency.

The cause of the inter-mission biases can in many cases be identified from difference plots during the mission overlap periods. In this study, we have identified the most dominating bending angle biases that are propagated from the input data or from the geophysical profile data to the gridded monthly means: lower-tropospheric biases linked to moist regions of the atmosphere, seasonally varying biases at high altitudes and high latitudes, Metop-COSMIC bias shifts related to firmware upgrades, and

a high-altitude hemispherically asymmetric bias related to small differences between the UCAR and EUMETSAT low-level input data. We also find systematic residual sampling errors that appear to be caused by the under-sampling of diurnal or semi-diurnal cycles not being fully corrected for by the sampling-error correction method.

The results presented here also affect the other geophysical variables retrieved from RO measurements, which are not explicitly discussed in the present study: dry pressure, dry geopotential heights, temperature, and humidity. For the latter two



variables, obtained through a 1D-Var retrieval using additional information from a model background (see Section 3.1), the relatively large inter-mission biases in the lower troposphere will have an impact on the temperature and humidity data records, which was not investigated here.

This study shows that above the lower troposphere and below about 30 km, data records from different RO satellite missions exhibit only small systematic differences. Further reduction of these differences most likely requires an improved sampling-error correction. Reducing the inter-mission differences at higher altitudes also requires reduced impacts from subtle differences in the input data, and from the statistical optimization of the bending angles, as well as an understanding of the cause of the high-altitude, high-latitude seasonally varying differences. A continued reduction of the relatively small, but systematic, inter-mission biases, is important for the generation of long-term stable, homogeneous RO-based CDRs extending to higher altitudes.

*Data availability.* The data used in the analysis are available at http://www.romsaf.org.

*Author contributions.* The data were generated as a team effort including all co-authors. Hans Gleisner performed the main analysis, generated the figures, and prepared the manuscript with contributions from the co-authors.

*Competing interests.* The authors declare that they have no conflict of interest.

*Acknowledgements.* This work was carried out as part of the Radio Occultation Meteorology Satellite Application Facility (ROM SAF),
which is a decentralised operational RO processing center under EUMETSAT. The authors are members of the ROM SAF team.





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
