# Peer review of "Evaluation of the 15-year ROM SAF monthly mean GPS radio occultation climate data record"

_Atmospheric Measurement Techniques, 2019_

## Referee Comment (RC1) · Richard Anthes (Referee) · 15 Dec 2019

**General comments**

This is a very well-written and informative paper that compares in detail the climate data record (CDR) of four radio occultation (RO) data sets: CHAMP, GRACE, COSMIC and Metop from 2001 to 2016. The four RO data sets are also compared with ERA-Interim (ERA-I) short-term forecasts. It is a valuable contribution to the literature documenting the characteristics of RO data sets. It shows the value of RO data for use in climate studies (particularly between 8 and 30 km) because of the overall accuracy and stability of RO from mission to mission. It also shows the value of RO missions (e.g. COSMIC) that fully sample the diurnal cycle.

[Figure]

The paper is acceptable for publication subject to consideration of a few minor changes, as summarized below:

Specific comments

1. It would be helpful if the paper would say a little more about the ERA-I data sets. On P2 line 35 to P3 line 1 it says "reanalysis data." Later, e.g. P4 line 2 it says "short-term forecasts." According to Berrisford et al. (2011 ECMWF Report), the ERA-I data sets include four analyses per day and two ten-day forecasts per day, initialized at 00 and 12 UTC. Exactly which of these data sets are being used to interpolate to the RO locations in space and time? Please define "short-term" and say why forecasts were used instead of analyses. Since the analysis fields are given only every six hours and the forecast fields are given every three hours, the higher temporal resolution of the forecast fields likely give better temporal interpolation to the RO times and this may be the reason why forecasts are used. A sentence or two would clarify exactly what is being done and why. A brief description of the temporal and horizontal and vertical interpolation schemes of ERA-I to RO locations would also be useful. 2. P5 lines 12-15-The paper mentions 1D-Var retrievals of water vapor, yet no water vapor comparisons of the RO data sets with themselves or with ERA-I are presented. Presumably this is outside the scope of this paper, but perhaps this should be stated here because I was expecting to see some water vapor comparisons later on. 3. The authors should state how ERA-I dry temperatures are computed for comparison with the RO dry temperatures, as in Fig. 6. Presumably the ERA dry temperature is computed by first computing the ERA-I refractivity N using the ERA-I temperature, pressure and water vapor pressure, and then computing ERA-I dry temperature from the Smith-Weintraub equation with e=0 (i.e. Tdry=77.6P/N). At first I thought this was RO dry temperatures compared with ERA-I temperatures, but this is clearly not the case. 4. P20 lines 6-7: The variation of the results with latitude bands compared to the global average is important. Most of the results in this paper are global means. The results are quite likely to be different in the tropics where the environment is more challenging for RO. Perhaps this

should be mentioned somewhere toward the front?

Technical/editorial comments

5. P2 line 4: "develops" should be "develop" 6. P2 line 24 and P26 line 3: "Rieck" should be "Rieckh". 7. P3 line 6: Please define "UCAR" (University Corporation for Atmospheric Research). 8. P9 line 8-"accuracy" should be "bias". 9. P13 line 6: better wording might be "..profiles deviate somewhat..." 10. P13 line 10: "lead" should be "leads" 11. P17 line 7: Suggested rewording rather than starting sentence with "Believed...": "This bias is believed to be...." or "We attribute this bias to the undersampling....." 12. P18 line 2: Suggested rewording rather than starting sentence with "Believed....":"This effect is believed to be..." or "We attribute this effect to the undersampling of the..." 13. P18 line 7: So far not Sofar

End of comments Richard Anthes

---

## Referee Comment (RC2) · Anonymous Referee #2 · 3 Feb 2020

This paper describes monthly gridded dataset obtained from GPS RO from multiple missions over a period of 15 years. This study quantifies the differences among the different missions as well as how they differ from ERA-interim analysis. The characterization of uncertainty is a necessary part of a climate data record, and this paper addresses mission differences arising from data quality and sampling. Overall, I think this paper presents some new and important results that as far as I know have not been documented previously. However, the paper can be improved by addressing the following comments:

(1) I would like to see more of the results broken down in different latitude bands, not just global averages. To minimize the number of figures, such information can perhaps be summarized in a table. After all, the CDR is a gridded dataset that covers

all latitudes, so users of this dataset would want to know how the results vary as a function of latitudes. (There are some mentions that the differences are up to a factor 2 larger for certain latitude bands but this is too vague.)

(2) Is the gridded dataset a 2D (latitude-height) or 3D (lat-lon-h)? Please specify up front.

(3) To provide a broader context for the readers, it would be useful to compare the multi-mission differences among the missions to the structural uncertainty estimated from the multi-center comparisons (e.g., Ho et al. 2009; Steiner et al. 2013).

(4) The paper mentions in multiple occasions that sampling error correction is necessary for combining the data from various RO missions. I don't necessarily agree. Wouldn't it be possible to combine the data and then perform the sampling error correction on the combined dataset?

(5) Section 2.1: Table 1 listed the input data used for the study. It would be helpful to provide some key information up front about the changes in versions since they can lead to abrupt changes in the time series. For example, we learned later on that the GRACE processing had changed from single-differencing to zero-differencing. Please also mention rationale for using the UCAR Metop input data even though it is not officially part of the CDR. Regarding the low-level input data, was there quality control done by UCAR and EUMETSAT at this point?

(6) Section 3.2: The results on the bending angle quality from different missions are interesting. However, for most readers not familiar to RO, it is not clear how they are relevant to the retrieved physical parameters shown later. Please provide some discussions of that.

(7) Figure 5: There is a significant jump between 2009-2010 between 20-30 km. Is that related to the "update of the COSMIC NRT data processing in October 2009"? Please provide more details.

(8) Figure 5: In 4-8 km height range, the inter-mission differences in bending angles seem big, but the refractivity differences appear to be much smaller (except for CHAMP). Why? Is it a matter of vertical scale in the plotting?

(9) Section 3.3: The authors describe the quality control criteria used to remove bad data. However, the descriptions were rather generic. For example, it's not clear how criteria a, b, and c are distinct from each other. I understand this is a complex subject and not the main focus of this paper, and the authors did reference a document on the ROM SAF web site. However, a little more technical information, along with some statistics on the percentage of occultations removed from each criterion, would be useful.

(10) Section 3.4: The authors describe the binning and averaging technique used to produce the gridded monthly means. It introduces a technique where each latitude bin is divided into two sub-bins to obtain latitude weighted average within each bin. What is the advantage of doing it this way vs. something like Gaussian weighting (for example)?

(11) Section 3.6: The authors stated that "In the generation of anomaly time series, the same seasonal cycle should be used for all missions and throughout the time series." I would argue that if you want to look at differences in seasonal cycles and anomalies separately, it would be better to derive the seasonal cycle from each time series and obtain the anomalies by removing corresponding seasonal cycles.

(12) Figure 5: There is a small constant bias (after 2006) between 8-20 km between RO and ERA-interim in both bending angle and refractivity. Can you comment on that?

(13) Section 5.1, p. 17: "Large-scale hemispherically asymmetric (north-south) Metop-COSMIC bias on the order of 0.1% above 35-40 km, and increasing upward... is believed to be related to differences in LEO satellite orbits from the two sources of input data." Can you provide some evidence of this claim?
(14) In general, for the bullets in pp. 15-17, the authors should use references when possible to substantiate the arguments.

(15) Section 5.2.1, p. 18: The difference between Metop and COSMIC bending angle over 8-30 km is large compared with other missions. Would that change if the Metop-UCAR input data were used instead?

(16) Also: "Metop shows . . . a stepwise decrease of the bias in mid-2013. . ." Any idea why?

(17) P. 2, line 4: Mission acronyms were never spelled out.

(18) P. 4, line 5: "The model data profiles are forward-modelled to the set of observed geophysical variables." Please provide more information on the forward modelling (e.g., from what variable to what variable, do you account for tangent point drifts, upper boundary for Abel integration, etc.).

(19) P. 18, line 6: Change "Sofar" to "So far"

(20) In the abstract and elsewhere, the authors differentiate the upper troposphere from lower by saying "6-8 km". I find this a little awkward (is it 6 or 8 km?). I think it's better to just use 8 km as the boundary.

---

## Author Comment (AC2) · 3 Mar 2020

**Author's reply to Referee Comment 2 (RC2)**

(1) Note that Figure 8 is actually broken down into latitudes. The latitude-resolved results in Figure 8 are discussed in Section 5.1. It is true that the vertically averaged time-series in Figs. 9 and 10 are only provided as global averages. The idea is that Figure 8 provides a detailed view across height and latitude for a limited set of RO missions and for bending angle only, while Figs. 9 and 10 provide a more complete view across RO missions and also for dry temperature. We will consider adding a table showing latitude-resolved vertically- and time-averaged RO mission differences as suggested by the reviewer. The exact contents of that table will have to be decided

after further considerations.

(2) The monthly mean data are provided on a 2D latitude-height grid. This is now mentioned up front in the Introduction: "... *we present results from an evaluation of the ROM SAF monthly-mean climate data records (CDRs) provided on a two-dimensional latitude-height grid, ...*"

(3) The multi-center inter-comparisons (Ho et al, 2009; Steiner et al., 2013) primarily resulted in quantifications of the trend differences between processing centers for RO variables retrieved from CHAMP data. Those results gave a first view of the uncertainties in the trends to be expected caused by differences between processing centers. The RO mission differences presented here constitute an additional source of uncertainty for the trends of long-term time series of RO data. However, we have only presented the differences themselves. We have not described the consequences for the long-term trends. We will come back to this issue in a future study, and by then we will hopefully be able to compare our results with an extended study of multi-center differences including several RO missions.

(4) Yes, it is possible to first combine the data and then do sampling error correction. This is actually our preferred method for constructing long multi-mission data records. However, it does not avoid the need to do sampling error correction as a remedy for differences in the sampling characteristics between RO missions. Either way (first do sampling error correction then combine data sets, or first combine data sets and then do sampling error correction) it is necessary to do sampling error correction if we want to avoid spurious long-term variability in the combined time series as new satellite missions replace older ones.

(5) We have updated the first paragraph of Section 2.1 accordingly:

*The ROM SAF CDR v1.0 includes data from four RO missions: CHAMP, GRACE, COSMIC, and Metop. The processing of data from the first three missions was based on low-level input data from UCAR, while the Metop data were processed with input*

*data from EUMETSAT. In addition, we also processed Metop data using input data from the University Corporation for Atmospheric Research (UCAR), to allow for investigation of differences related to the low-level input data. The low-level input data consist of amplitude and excess phase data, together with positions and velocities for Global Positioning System (GPS) and low-Earth orbit (LEO) satellites. The input data versions are shown in Table 1. During the time period, COSMIC and GRACE experienced version updates, where the latter included a switch to zero-differencing in the generation of excess phases.*

Concerning the question of whether there is quality control done on the low-level data, we know that EUMETSAT does it and it is actually taken into account in our own QC. Regarding UCAR data, we would assume there is a certain degree of low-level quality control and data rejection. However, it is difficult to find detailed documentation on that, and it is also a bit out of the scope of this article to go into such detail.

(6) We have added a new second paragraph to Section 3.2:

*High noise levels and other errors may lead to occultations being rejected by the quality screening. Any errors in the retained bending angles may propagate further down the processing chain, since bending angles are the starting point for the retrieval of the other geophysical variables. In particular, bending angle data in the upper stratosphere is affected by residual ionospheric errors resulting in errors in refractivity and dry temperature lower down in the stratosphere.*

(7) The jump that we see in the RO-ERAI differences (Figure 5) in October 2009 is related to a bias shift in ERA-Interim. It is our understanding that this bias shift in ERA-Interim was caused by an update of the COSMIC data processing at UCAR, leading to a change in the data being assimilated by the ERA-Interim reanalysis system. The reference provided (Healy, 2013) states that the change in the operational processing of COSMIC data took place on October 12, 2009, which is consistent with what we see in our data records.

(8) The inter-mission differences in the 4-8 km height range are actually larger for bending angle than for refractivity. The effect can also be seen in Figure 6 for time-averaged data. It is partly a matter of the vertical scale of the plots: the bending angle is plotted as a function of impact altitude while the refractivity is plotted versus altitude. The two vertical height variables can differ by up to 1-2 km near the surface. The downward propagation of information through the Abel transform also means that at a particular height (altitude or impact altitude) in the 4-8 km height range, less biased data from higher atmospheric layers contribute a larger fraction of the data for the refractivity compared to bending angle.

(9) We have now updated the text in Section 3.3 in order to provide somewhat more information:

*Before processing the atmospheric profiles to gridded monthly-mean data, all profiles are checked against a set of quality criteria. The quality criteria include tests to identify occultations that a) do not provide any meaningful bending angles or only provide bending angles in a limited height range, b) have degraded bending angles indicated by increased noise in the L2 signal or by certain types of deviation from a bending-angle climatology, c) could be regarded as outliers as evidenced by comparison with ECMWF reanalysis data, or d) encounter problems in the 1D-Var processing. More detailed descriptions of the data quality screening are found in the series of validation reports available at the ROM SAF web site (http://www.romsaf.org/product_documents.php).*

*If an occultation does not pass one or several of the tests in a, b, or c, the bending angle, refractivity, and dry variables are marked as non-nominal. Otherwise, they are regarded as nominal and the refractivity profiles are passed on to the 1D-Var processing. The fraction of data rejected in the quality screening varies with time (Fig. 1). On average, around 10-20% of the occultations are rejected, although with large differences between the RO missions. Metop and GRACE show the highest throughput of data; almost no data are rejected by criterion a and about 5-10% are rejected by criteria b and c. COSMIC and CHAMP have roughly similar overall rejection rates. However,*

*for COSMIC about 5-10% of data are rejected by criterion a, while for CHAMP that criterion removes about 15% or even more of the data.*

(10) The purpose of the division into sub-bins is to compute a bin mean that as closely as possible approximate an area-weighted mean:

$$\bar{X} = \frac{1}{A} \int X dA \qquad (1)$$

where $X$ is a geophysical variable and $A$ is the bin area. The main problem that is addressed by the weighting is non-uniform sampling of $X$ across the bin. As stated in the text, we only consider the non-uniform distribution of observations in latitude. The division of the 5-degree latitude bins into two sub-bins can be regarded as a very coarse discretization of the above mentioned integral. During time periods with a lot of RO data, and for monthly bins, a finer discretization would be possible. However, we want to use the same binning across the whole time series, from CHAMP to COSMIC and Metop, and, hence, we chose two sub-bins for the 5-degree main bins.

(11) First a clarification: as stated in Section 3.6, by the "mean seasonal cycle" we mean the long-term average as a function of latitude, height, and season. For each latitude, height, and January month, we average across many years. Similarly for all February months, etc. This is what Eq. 10 expresses. Our "mean seasonal cycle" is a sum of the long-term means (i.e. means across all seasons) and the seasonal cycle on top of that long-term mean. We do not separate the two.

Let's say that we have two overlapping missions, A and B. Should we compute anomalies for A using data from A as reference, and anomalies for B using B data as reference, we wouldn't be able to detect constant biases between the missions. The same argument could be made even if we separate long-term means from the seasonal cycle: we wouldn't be able to detect differences in the seasonal cycle as measured by A and by B.

In addition, assume that we generate long-term climatologies from several non-overlapping RO missions. If we construct the anomalies using mission-specific anomaly references, we would not only remove any biases (systematic errors) between the missions, but also true climatological time variations.

(12) In Section 4, we now mention that even though the magnitude of the RO-ERAI difference suddenly decreased as a result of the start of assimilation of COSMIC data in 2006, the difference remains slightly negative in the 12-20 km height interval throughout the data record.

(13) In our input data we find differences between the EUMETSAT and UCAR LEO orbits with the right magnitude, and with a periodicity of one orbit, that potentially could explain the hemispheric differences that we observe (see Figure 1). The source, and the ultimate cause, of these orbital differences is still a matter of ongoing work. More detailed information can also be found in the series of validation reports available at http://www.romsaf.org/product_documents.php. It is, however, still premature and out of the scope of this paper to include a detailed description of what we believe is the cause of the differences.

(14) The purpose of the bullet list is to summarize our own observations of inter-mission biases as shown in Figure 8 (and to some extent Figure 9). In some bullets we also point out possible causal relations, a few of them well-known while other should be regarded as preliminary conclusions from the present study itself. We will add references to the first bullet (signal tracking issues) and the third bullet (Metop software upgrades).

(15) No, the Metop-COSMIC globally averaged differences that we see in Figure 9 would be very similar for Metop(UCAR) as for Metop(EUM). You can see this in Figure 8, by comparing the middle column (Metop(UCAR) minus COSMIC) with the right-most column (Metop(EUM) minus COSMIC). The dominating differences between Metop(UCAR) and Metop(EUM) are a large-scale hemispheric asymmetry (see the top panels in Figure 8) and some differences in the lower troposphere.

(16) The Metop-COSMIC bias shift in 2013 can also be identified in Figure 8, where you also can see the latitude distribution of the shift. It is related to firmware upgrades in the Metop RO instruments described by the third bullet in Section 5.1.

(17) Mission acronyms are now spelled out in the Introduction.

(18) We have now expanded Section 2.2 somewhat, partly based on the comments of another reviewer. We also added a new reference (Healy and Thépaut, 2006) that provides a detailed description of the forward-modelling to refractivity and bending angle.

*We used ERA-Interim reanalysis (Dee et al., 2011) data as a reference in the evaluation. To avoid the direct impact of the observed data on our comparison reference (RO data are assimilated by ERA-Interim), we used the reanalysis forecasts rather than analyses. ERA-Interim provides forecasts at three-hour intervals, intialized at 00 and 12 UTC. Hence, the shortest possible forecast time vary from 3 hours to 12 hours. For each RO event, a co-located vertical profile of model data was obtained by interpolation in the global forecast fields representing the atmospheric state at three-hour intervals (UTC 00, 03, . . .) on a 1.0 x 1.0 latitude-longitude grid.*

*The vertical profiles of model data (pressure, temperature, and humidity as function of geopotential height) are forward-modelled to the set of geophysical variables used in this study. The model refractivity is calculated from the Smith-Weintraub equation, and the bending angles are obtained by an Abel integral over the refractivity profile assuming an exponential decay above the model top (Healy and Thépaut, 2006). Dry temperature profiles are computed from the model refractivities using the same method as for the observed profiles (see Section 3.1). This is followed by monthly averaging in latitude bins and interpolation onto an equidistant 200 meter height grid, using the methods described in Section 3.4.*

(19) Section 5.2 now begin with: "*The RO mission differences have so far been described . . .*".

(20) The notation "6-8 km" was used to indicate that this is not an exact altitude. We have now changed four occurrences of "6-8 km" in the manuscript to "8 km" (in Abstract and in Sections 1 and 5.1).

[Figure]

**Fig. 1.** Differences between Metop orbits from EUMETSAT and UCAR, as quantified by the differences in the Metop-GPS distances.

---

## Author Response (AR1)

**Author's final comments and changes in the manuscript _amt-2019-417_**

**Reviewer #1, specific comments**

**(1)**
_Reviewer:_ It would be helpful if the paper would say a little more about the ERA-I data sets. On P2 line 35 to P3 line 1 it says "reanalysis data." Later, e.g. P4 line 2 it says "short-term forecasts." According to Berrisford et al. (2011 ECMWF Report), the ERA-I data sets include four analyses per day and two ten-day forecasts per day, initialized at 00 and 12 UTC. Exactly which of these data sets are being used to interpolate to the RO locations in space and time? Please define "short-term" and say why forecasts were used instead of analyses. Since the analysis fields are given only every six hours and the forecast fields are given every three hours, the higher temporal resolution of the forecast fields likely give better temporal interpolation to the RO times and this may be the reason why forecasts are used. A sentence or two would clarify exactly what is being done and why. A brief description of the temporal and horizontal and vertical interpolation schemes of ERA-I to RO locations would also be useful.

_Author response:_ A problem with using reanalysis data as a reference for comparison with radio occultation (RO) data is that RO data itself have been assimilated by the reanalysis system. If we interpolate in an ERA-I _analysis_ field to the times and locations of RO events, the interpolated model data will be strongly influenced by the very same RO data that we wish to compare with. Such comparisons would have an obvious problem with circularity. Using ERA-I _forecasts_ instead of _analyses_ removes most of this circularity. ERA-I provides forecasts at 3-hour intervals, initialized at 00 and 12 UTC. The shortest possible forecast times thus vary from 3 hours up to 12 hours. This is what we mean by "short-term" forecasts.

The model resolution is 3 hours and data are interpolated from a 1.0x1.0 degree latitude-longitude grid. In the vertical, as a part of the forward-modelling from model space to observation space, the data are first interpolated from the original ECMWF model levels to the vertical coordinates of the observed data. All data – both observed and model data - are then interpolated to a regular 200-meter grid that is used in this study.

_Manuscript changes:_ We have extended Section 2.2, which now reads: "_We used ERA-Interim reanalysis \citep{DeeEtal2011} data as a reference in the evaluation. To avoid the direct impact of the observed data on our comparison reference (RO data are assimilated by ERA-Interim), we used the reanalysis forecasts rather than analyses. ERA-Interim provides forecasts at three-hour intervals, intialized at 00 and 12 UTC. Hence, the shortest possible forecast time vary from 3 hours to 12 hours. For each RO event, a co-located vertical profile of model data was obtained by interpolation in the global forecast fields representing the atmospheric state at three-hour intervals (UTC 00, 03, \ldots) on a $1.0^{\circ}\times1.0^{\circ}$ latitude-longitude grid. The model data are forward-modelled to the set of geophysical variables used in this study. Dry temperature profiles are computed from the ERA-Interim refractivities using the same method as for the observed profiles (see Section~3.1). This is followed by monthly averaging in latitude bins and interpolation onto an equidistant 200 meter height grid, using the methods described in Section~3.4_".

**(2)**

*Reviewer:* P5 lines 12-15. Thepaper mentions 1D-Var retrievals of water vapor, yet no water vapor comparisons of the RO data sets with themselves or with ERA-I are presented. Presumably this is outside the scope of this paper, but perhaps this should be stated here because I was expecting to see some water vapor comparisons later on.

*Author response:* The 1D-Var retrievals are mentioned for completeness. From a climate perspective, the 1D-Var retrievals come with their own set of problems that are only partly related to the more fundamental errors and uncertainties of the bending-angle, refractivity, and dry temperature data. A discussion of the problems and errors with the 1D-Var retrievals is out of the scope of this paper. We will mention this in Section 3.1 of the updated manuscript by adding a sentence.

*Manuscript changes:* Added a sentence to Section 3.1: "*The tropospheric variables retrieved through a 1D-Var algorithm are not discussed further in this paper*".

**(3)**

*Reviewer:* The authors should state how ERA-I dry temperatures are computed for comparison with the RO dry temperatures, as in Fig. 6. Presumably the ERA dry temperature is computed by first computing the ERA-I refractivity N using the ERA-I temperature, pressure and water vapor pressure, and then computing ERA-I dry temperature from the Smith-Weintraub equation with e=0 (i.e., Tdry=77.6P/N). At first I thought this was RO dry temperatures compared with ERA-I temperatures, but this is clearly not the case.

*Author response:* The ERA-I dry temperatures are computed by first forward modelling ERA-I pressure, temperature, and humidity to ERA-I refractivity. Then we use similar processing as for the observed refractivity: we assume hydrostatic equilibrium and do a downward vertical integration of the hydrostatic integral from the top of the atmosphere to get the dry pressure. The refractivity and the dry pressure then give us the dry temperature under the assumption of zero water vapour pressure. We will update the text in Section 2.2 to be more specific on this – see suggested text update under point 1 above.

*Manuscript changes:* see under (1) above.

**(4)**

*Reviewer:* P20 lines 6-7: The variation of the results with latitude bands compared to the global average is important. Most of the results in this paper are global means. The results are quite likely to be different in the tropics where the environment is more challenging for RO. Perhaps this should be mentioned somewhere toward the front?

*Author response:* Yes, many of the results in the paper are global means, as we had to limit the discussions somehow. However, Figure 8, which is a key figure in this paper, gives a quite detailed view of the latitudinal differences. These are discussed in some detail in Section 5.1. There are indications both in the Abstract and in Section 5.2 that the consistency amongst RO missions is lower for limited latitude bands than for global means.

It is true that the environment is challenging for RO in the tropics, particularly in the lower and middle troposphere where humidity has a large impact on the refractivity. In general, the lower troposphere is challenging, and this is also where we find the largest biases between the RO missions. The main limitation of the study is actually in the vertical. The large differences between the RO missions below 8 km are only mentioned briefly while the differences above 8 km are discussed in some detail. We will add a sentence in the Introduction to point out this limitation.

*Manuscript changes:* Sentence added in the Introduction: "*The evaluation is largely limited to the stratosphere and the upper troposphere, above about 6-8 km*".

**Reviewer #1, editorial/technical comments**

**(5)** P2 line 4: "develops" should be "develop"
**(6)** P2 line 24 and P26 line 3: "Rieck" should be "Rieckh".
**(7)** P3 line 6: Please define "UCAR" (University Corporation for Atmospheric Research).
**(8)** P9 line 8-"accuracy" should be "bias".
**(9)** P13 line 6: better wording might be "..profiles deviate somewhat. . ."
**(10)** P13 line 10: "lead" should be "leads"
**(11)** P17 line 7: Suggested rewording rather than starting sentence with "Believed. . .": "This bias is believed to be. . .." or "We attribute this bias to the undersampling.... . ."
**(12)** P18 line 2: Suggested rewording rather than starting sentence with "Believed. . ..":"This effect is believed to be. . ." or "We attribute this effect to the undersampling of the. . ."
**(13)** P18 line 7: So far not Sofar

*Manuscript changes:* Text updated accordingly.

**Reviewer #2, specific comments**

**(1)**
*Reviewer:* I would like to see more of the results broken down in different latitude bands, not just global averages. To minimize the number of figures, such information can perhaps be summarized in a table. After all, the CDR is a gridded dataset that covers all latitudes, so users of this dataset would want to know how the results vary as a function of latitudes. (There are some mentions that the differences are up to a factor 2 larger for certain latitude bands but this is too vague.)

*Author response:* Note that Figure 8 is actually broken down into latitudes. The latitude-resolved results in Figure 8 are discussed in Section 5.1. It is true that the vertically averaged time-series in Figs. 9, 10, and 11 are only provided as global averages. The idea is that Figure 8 provides a detailed view across height and latitude for a limited set of RO missions and for bending angle only, while Figs. 9, 10, and 11 provide a more complete view across RO missions and for all three geophysical variables. Anyway, we have now added a table showing latitude-resolved time-averaged RO mission differences as suggested by the reviewer.

*Manuscript changes:* Added a Table 2 and references to it in the end of Sections 5.2.1, 5.2.2, and 5.2.3.

**(2)**
*Reviewer:* Is the gridded dataset a 2D (latitude-height) or 3D (lat-lon-height)? Please specify up front.

*Author response:*  The monthly mean data are provided on a 2D latitude-height grid. This is now mentioned in the Introduction.

*Manuscript changes:* Added to the Introduction: "…. *we present results from an evaluation of the ROM SAF monthly-mean climate data records (CDRs) provided on a two-dimensional latitude-height grid, ….*".

**(3)**
*Reviewer:* To provide a broader context for the readers, it would be useful to compare the multi-mission differences among the missions to the structural uncertainty estimated from the multi-center comparisons (e.g., Ho et al. 2009; Steiner et al. 2013).

*Author response:* The multi-center inter-comparisons (Ho et al, 2009; Steiner et al., 2013) primarily resulted in quantifications of the trends in the differences between processing centers for RO variables retrieved from CHAMP data. Those results gave a first view of the uncertainties in the trends to be expected from the differences between processing centers (limited to CHAMP data only). The RO mission differences presented here constitute an additional source of uncertainty for the trends of long-term time series of RO data. However, we have only presented the differences themselves. We have not described the consequences for the long-term trends. We will come back to this issue in a future study, and by then we will hopefully be able to compare our results with an extended study of multi-center differences including several RO missions.

**(4)**

*Reviewer:* The paper mentions in multiple occasions that sampling error correction is necessary for combining the data from various RO missions. I don't necessarily agree. Wouldn't it be possible to combine the data and then perform the sampling error correction on the combined dataset?

*Author response:* Yes, it is possible to first combine the data and then do sampling error correction. This is actually our preferred method for constructing long multi-mission data records. However, it does not avoid the need to do sampling error correction as a remedy for differences in the sampling characteristics between RO missions. Either way (first do sampling error correction then combine data sets, or first combine data sets and then do sampling error correction) it is necessary to do sampling error correction if we want to avoid spurious long-term variability in the combined time series as new satellite missions replace older ones.

**(5)**

*Reviewer:* Section 2.1: Table 1 listed the input data used for the study. It would be helpful to provide some key information up front about the changes in versions since they can lead to abrupt changes in the time series. For example, we learned later on that the GRACE processing had changed from single-differencing to zero-differencing. Please also mention rationale for using the UCAR Metop input data even though it is not officially part of the CDR. Regarding the low-level input data, was there quality control done by UCAR and EUMETSAT at this point?

*Author response:* We have updated the first paragraph of Section 2.1 accordingly. The statement about the change from single- to zero-differencing was removed. This statement was based on a) the paper *Angerer et al.* (2017) where it was stated that UCAR used single-differencing for GRACE data version 2010.2640, and b) it was consistent with what we see in our time series (e.g., Figure 2). However, from personal communication with D. Hunt at UCAR we have been informed that all GRACE data from UCAR are based on zero-differencing. Concerning the question of whether there is quality control done on the low-level data, we know that EUMETSAT does it and it is actually taken into account in our own QC. Regarding UCAR data, we would assume there is a certain degree of low-level quality control and data rejection. However, it is difficult to find detailed documentation on that, and it is also a bit out of the scope of this article to go into such detail.

*Manuscript changes:* The first paragraph of Section 2.1 now reads: "*The ROM SAF CDR v1.0 includes data from four RO missions: CHAMP, GRACE, COSMIC, and Metop. The processing of data from the first three missions was based on low-level input data from the University Corporation for Atmospheric Research (UCAR), while the Metop data were processed with input data from EUMETSAT. In addition, we also processed Metop data using input data from UCAR, to allow for investigation of differences related to the low-level input data. The low-level input data consist of amplitude and excess phase data, together with positions and velocities for Global Positioning System (GPS) and low-Earth orbit (LEO) satellites. The input data versions are shown in Table 1. During the time period, COSMIC and GRACE experienced version updates that involved low-level processing software changes at UCAR*".

**(6)**

*Reviewer:* Section 3.2: The results on the bending angle quality from different missions are interesting. However, for most readers not familiar to RO, it is not clear how they are relevant to the retrieved physical parameters shown later. Please provide some discussions of that.

*Author response:* We have added a new second paragraph to Section 3.2.

*Manuscript changes:* A new, second, paragraph was added to Section 3.2: "*High noise levels and other errors may lead to occultations being rejected by the quality screening. Any errors in the retained bending angles may propagate further down the processing chain, since bending angles are the starting point for the retrieval of the other geophysical variables. In particular, bending angle data in the upper stratosphere is affected by residual ionospheric errors resulting in errors in refractivity and dry temperature lower down in the stratosphere*". In addition, the last sentence of the fourth paragraph was slightly changed. It now reads: "*This stepwise change is related to a switch of input data version (Table 1) involving several changes in UCAR's processing software (D. Hunt, pers. comm.)*".

**(7)**

*Reviewer:* Figure 5: There is a significant jump between 2009-2010 between 20-30 km. Is that related to the "update of the COSMIC NRT data processing in October 2009"? Please provide more details.

*Author response:* The jump that we see in the RO-ERAI differences (Figure 5) in October 2009 is related to a bias shift in ERA-Interim. It is our understanding that this bias shift in ERA-Interim was caused by an update of the COSMIC data processing at UCAR, leading to a change in the data being assimilated by the ERA-Interim reanalysis system. The reference provided (Healy, 2013) states that the change in the operational processing of COSMIC data took place on October 12, 2009, which is consistent with what we see in our data records.

*Manuscript changes:* Minor change of a sentence in the second paragraph of Section 4 to "… *an update of the COSMIC data processing in October 2009 (Healy, 2013), …*".

**(8)**

*Reviewer:* Figure 5: In 4-8 km height range, the inter-mission differences in bending angles seem big, but the refractivity differences appear to be much smaller (except for CHAMP). Why? Is it a matter of vertical scale in the plotting?

*Author response:* The inter-mission differences in the 4-8 km height range are actually larger for bending angle than for refractivity. The effect can also be seen in Figure 6 for time-averaged data. It is partly a matter of the vertical scale of the plots: the bending angle is plotted as a function of impact altitude while the refractivity is plotted versus altitude. The two vertical height variables can differ by up to 1-2 km near the surface. The downward propagation of information through the Abel transform also means that at a particular height (altitude or impact altitude) in the 4-8 km height range, less biased data from higher atmospheric layers contribute a larger fraction of the data for the refractivity compared to bending angle.

**(9)**

**Reviewer:** Section 3.3: The authors describe the quality control criteria used to remove bad data. However, the descriptions were rather generic. For example, it's not clear how criteria a, b, and c are distinct from each other. I understand this is a complex subject and not the main focus of this paper, and the authors did reference a document on the ROM SAF web site. However, a little more technical information, along with some statistics on the percentage of occultations removed from each criterion, would be useful.

**Author response:** We have now updated the text in Section 3.3 in order to provide some more information.

**Manuscript changes:** The text in Section 3.3 has been updated and now reads: *"Before processing the atmospheric profiles to gridded monthly-mean data, all profiles are checked against a set of quality criteria. The quality criteria include tests to identify occultations that a) do not provide any meaningful bending angles or only provide bending angles in a limited height range, b) have degraded bending angles indicated by increased noise in the L2 signal or by certain types of deviation from a bending-angle climatology, c) could be regarded as outliers as evidenced by comparison with ECMWF reanalysis data, or d) encounter problems in the 1D-Var processing. More detailed descriptions of the data quality screening are found in the series of validation reports available at the ROM SAF web site*
*(http://www.romsaf.org/product_documents.php).*

*If an occultation does not pass one or several of the tests in a, b, or c, the bending angle, refractivity, and dry variables are marked as non-nominal. Otherwise, they are regarded as nominal and the refractivity profiles are passed on to the 1D-Var processing. The fraction of data rejected in the quality screening varies with time (Fig. 1). On average, around 10-20 % of the occultations are rejected, although with large differences between the RO missions. Metop and GRACE show the highest throughput of data; almost no data are rejected by criterion a and about 5-10 % are rejected by criteria b and c. COSMIC and CHAMP have roughly similar overall rejection rates. However, for COSMIC about 5-10 % of data are rejected by criterion a, while for CHAMP that criterion removes about 15 % or even more of the data".*

**(10)**

**Reviewer:** Section 3.4: The authors describe the binning and averaging technique used to produce the gridded monthly means. It introduces a technique where each latitude bin is divided into two sub-bins to obtain latitude weighted average within each bin. What is the advantage of doing it this way vs. something like Gaussian weighting (for example)?

**Author response:** The purpose of the division into sub-bins is to compute a bin mean that as closely as possible approximate an area-weighted mean:

$$\bar{X} = \frac{1}{A}\int XdA$$

where *X* is a geophysical variable and *A* is the bin area. The main problem that is addressed by the weighting is non-uniform sampling of *X* across the bin. As stated in the text, we only consider the non-uniform distribution of observations in latitude. The division of the 5-degree latitude bins into two sub-bins can be regarded as a very coarse discretization of the above mentioned integral. During time periods with a lot of RO data, and for monthly bins, a finer discretization would be possible. However, we want to use the same binning across the whole time series, from CHAMP to COSMIC and Metop, and, hence, we chose two sub-bins for the 5-degree main bins.

**(11)**

***Reviewer:*** Section 3.6: The authors stated that "In the generation of anomaly time series, the same seasonal cycle should be used for all missions and throughout the time series." I would argue that if you want to look at differences in seasonal cycles and anomalies separately, it would be better to derive the seasonal cycle from each time series and obtain the anomalies by removing corresponding seasonal cycles.

***Author response:*** First a clarification: as stated in Section 3.6, by the "mean seasonal cycle" we mean the long-term average as a function of latitude, height, and season. For each latitude, height, and January month, we average across many years. Similarly for all February months, etc. This is what Eq. 10 expresses. Our "mean seasonal cycle" is a sum of the long-term all-season means and the seasonal cycle on top of that long-term mean. We do not separate the two.

Let's say that we have two overlapping missions, A and B. Should we compute anomalies for A using data from A as reference, and anomalies for B using data from B as reference, we wouldn't be able to detect constant biases between the missions. The same argument could be made even if we separate long-term all-season means from the seasonal cycle: we wouldn't be able to detect differences in the seasonal cycle as measured by A and by B.

In addition, assume that we generate long-term climatologies from several non-overlapping RO missions. If we construct the anomalies using mission-specific anomaly references, we would not only remove any biases (systematic errors) between the missions, but also true climatological time variations.

**(12)**

***Reviewer:*** Figure 5: There is a small constant bias (after 2006) between 8-20 km between RO and ERA-interim in both bending angle and refractivity. Can you comment on that?

***Author response:*** In Section 4, we now mention that even though the magnitude of the RO-ERAI difference suddenly decreased as a result of the start of assimilation of COSMIC data in 2006, the difference remains slightly negative in the 12-20 km height interval throughout the data record.

***Manuscript changes:*** The updated sentence in the second paragraph of Section 4 now reads: *"In December 2006 the magnitude of the bias suddenly decreased in the 12-20 km and 30-40 km intervals, even though it remained slightly negative in the 12-20 km range throughout the time period"*.

**(13)**

*Reviewer:* Section 5.1, p. 17: "Large-scale hemispherically asymmetric (north-south) Metop-COSMIC bias on the order of 0.1% above 35-40 km, and increasing upward . . . is believed to be related to differences in LEO satellite orbits from the two sources of input data." Can you provide some evidence of this claim?

*Author response:* In our input data we find differences between the EUMETSAT and UCAR LEO orbits with the right magnitude, and with a periodicity of one orbit, that potentially could explain the hemispheric differences that we observe (see Figure A below). The source, and the ultimate cause, of these orbital differences is still a matter of ongoing work. More detailed information can also be found in the series of validation reports available at http://www.romsaf.org/product_documents.php. It is, however, still premature and out of the scope of this paper to include a detailed description of what we believe is the cause of the differences.

[Figure]

***Figure A.*** *Differences between Metop orbits from EUMETSAT and UCAR, as quantified by the differences in the Metop-GPS distances.*

**(14)**

*Reviewer:* In general, for the bullets in pp. 15-17, the authors should use references when possible to substantiate the arguments.

*Author response:* The purpose of the bullet list is to summarize our own observations of inter-mission biases as shown in Figure 8 (and to some extent Figure 9). In some bullets we also point out possible causal relations, a few of them well-known while other should be regarded as preliminary conclusions from the present study itself. We have added references to the first bullet (signal tracking issues) and the third bullet (Metop software upgrades).

*Manuscript changes:* First bullet in Section 5.1 now ends with: "… the atmosphere (e.g., *Sokolovskiy et al.,* 2010)". Third bullet ends with: "… appeared (described in the series of ROM SAF validation reports at *http://www.romsaf.org/product_documents.php*)".

**(15)**

*Reviewer:* Section 5.2.1, p. 18: The difference between Metop and COSMIC bending angle over 8-30 km is large compared with other missions. Would that change if the Metop-UCAR input data were used instead?

*Author response:* No, the Metop-COSMIC globally averaged differences that we see in Figure 9 would be very similar for Metop(UCAR) as for Metop(EUM). You can see this in Figure 8, by comparing the middle column (Metop(UCAR) minus COSMIC) with the rightmost column (Metop(EUM) minus COSMIC). The dominating differences between Metop(UCAR) and Metop(EUM) are a large-scale hemispheric asymmetry (see the top panels in Figure 8) and some differences in the lower troposphere.

**(16)**

*Reviewer:* Also: "Metop shows . . . a stepwise decrease of the bias in mid-2013. . ." Any idea why?

*Author response:* The Metop-COSMIC bias shift in 2013 can also be identified in Figure 8, where you also can see the latitude distribution of the shift. It is related to firmware upgrades in the Metop RO instruments described by the third bullet in Section 5.1.

**(17)**

*Reviewer:* P. 2, line 4: Mission acronyms were never spelled out.

*Author response:* Mission acronyms are now spelled out in the Introduction.

*Manuscript changes:* The second paragraph of Section 1 now begins: "The RO Meteorology Satellite Application Facility (ROM SAF), which is a decentralized operational RO processing center under EUMETSAT, has recently undertaken a reprocessing of RO data from four satellite missions: CHAMP (CHAllenging Minisatellite Payload; *Wickert et al.*, 2001), GRACE (Gravity Recovery and Climate Experiment; *Beyerle et al.*, 2005), COSMIC (Constellation Observing System for Meteorology, Ionosphere, and Climate; *Anthes et al.*, 2008), and Metop (*Luntama et al.*, 2008)".

**(18)**
**Reviewer:** P. 4, line 5: "The model data profiles are forward-modelled to the set of observed geophysical variables." Please provide more information on the forward modelling (e.g., from what variable to what variable, do you account for tangent point drifts, upper boundary for Abel integration, etc.).

**Author response:** We have now expanded Section 2.2 somewhat, based on the comments of both reviewers. We also added a new reference (*Healy and Thépaut*, 2006) that provides a detailed description of the forward-modelling to refractivity and bending angle.

**Manuscript changes:** The updated Section 2.2 now reads: *"We used ERA-Interim reanalysis (Dee et al., 2011) data as a reference in the evaluation. To avoid the direct impact of the observed data on our comparison reference (RO data are assimilated by ERA-Interim), we used the reanalysis forecasts rather than analyses. ERA-Interim provides forecasts at three-hour intervals, intialized at 00 and 12 UTC. Hence, the shortest possible forecast time vary from 3 hours to 12 hours. For each RO event, a co-located vertical profile of model data was obtained by interpolation in the global forecast fields representing the atmospheric state at three-hour intervals (UTC 00, 03, ...) on a $1.0^O$ x $1.0^O$ latitude-longitude grid.*

*The vertical profiles of model data (pressure, temperature, and humidity as function of geopotential height) are forward-modelled to the set of geophysical variables used in this study. The model refractivity is calculated from the Smith-Weintraub equation, and the bending angles are obtained by an Abel integral over the refractivity profile assuming an exponential decay above the model top (Healy and Thépaut, 2006). Dry temperature profiles are computed from the model refractivities using the same method as for the observed profiles (see Section 3.1). This is followed by monthly averaging in latitude bins and interpolation onto an equidistant 200 meter height grid, using the methods described in Section 3.4".*

**(19)**
**Reviewer:** P. 18, line 6: Change "Sofar" to "So far"

**Author response:** Done.

**Manuscript changes:** Section 5.2 now begin: "*The RO mission differences have so far been described ....*".

**(20)**
**Reviewer:** In the abstract and elsewhere, the authors differentiate the upper troposphere from lower by saying "6-8 km". I find this a little awkward (is it 6 or 8 km?). I think it's better to just use 8 km as the boundary.

**Author response:** The notation "6-8 km" was used to indicate that this is not an exact altitude. We have now changed four occurrences of "6-8 km" in the manuscript to "8 km" (in Abstract and in Sections 1 and 5.1).

**Manuscript changes:** Changed four occurrences of "6-8 km" to "8 km" in Abstract and in Sections 1 and 5.1.

[revised manuscript text omitted]

---

## Author Response (AR2)

**Author's reply to Editors final comment**

*Editor:* I have just a minor comment before publication. Please explain your procedure in relation to Comment (4) of Reviewer #2. Before or after combining the data ? Is before a better option ? Why ?

[ *Reviewer #2, comment 4:* The paper mentions in multiple occasions that sampling error correction is necessary for combining the data from various RO missions. I don't necessarily agree. Wouldn't it be possible to combine the data and then perform the sampling error correction on the combined dataset? ]

*Author response:* Our response to comment 4 from Reviewer #2 was that we believe sampling error correction is a necessary step, and we explained that the order the reviewer suggests (first average all relevant data, then do sampling error correction) is actually the way we currently do it within the ROM SAF when combing data from several RO missions. Averaging of all relevant RO data before sampling error correction is justified if the systematic errors are small, and substantially simplifies the handling of differences in the sampling characteristics.

However, note that we do not directly study multi-mission data records in the present manuscript. We only study differences between single-mission data records. Anyway, we have now updated the manuscript to be a bit more precise on our averaging technique.

[revised manuscript text omitted]